# Nanobody-based RFP-dependent Cre recombinase for selective anterograde tracing in RFP-expressing transgenic animals

Ayumu Inutsuka [1✉], Sho Maejima[2], Hiroyuki Mizoguchi[3], Ryosuke Kaneko[4], Rei Nomura[2], Keiko Takanami [2], Hirotaka Sakamoto [2] & Tatsushi Onaka [1✉]

Transgenic animals expressing fluorescent proteins are widely used to label specific cells and proteins. By using a split Cre recombinase fused with mCherry-binding nanobodies or designed ankyrin repeat proteins, we created Cre recombinase dependent on red fluorescent protein (RFP) (Cre-DOR). Functional binding units for monomeric RFPs are different from those for polymeric RFPs. We confirmed selective target RFP-dependent gene expression in the mouse cerebral cortex using stereotaxic injection of adeno-associated virus vectors. In estrogen receptor-beta (*Esr2*)-mRFP1 mice and gastrin-releasing peptide receptor (*Grpr*)-mRFP1 rats, we confirmed that Cre-DOR can be used for selective tracing of the neural projection from RFP-expressing specific neurons. Cellular localization of RFPs affects recombination efficiency of Cre-DOR, and light and chemical-induced nuclear translocation of an RFP-fused protein can modulate Cre-DOR efficiency. Our results provide a method for manipulating gene expression in specific cells expressing RFPs and expand the repertoire of nanobody-based genetic tools.

[1] Division of Brain and Neurophysiology, Department of Physiology, Jichi Medical University, Shimotsuke, Tochigi 329-0498, Japan. [2] Ushimado Marine Institute (UMI), Graduate School of Natural Science and Technology, Okayama University, Ushimado, Setouchi, Okayama 701-4303, Japan. [3] Department of Neuropsychopharmacology and Hospital Pharmacy, Nagoya University Graduate School of Medicine, Nagoya, Aichi 466-8560, Japan. [4] KOKORO-Biology Group, Laboratories for Integrated Biology, Graduate School of Frontier Biosciences, Osaka University, Suita, Osaka 565-0871, Japan. ✉email: inutsuka@jichi.ac.jp; tonaka@jichi.ac.jp

To understand the precise roles of specific types of neurons, their visualization and manipulation in a living animal are required. Genetically encoded fluorescent proteins are widely used to identify and observe specific types of cells in vivo. There are various fluorescent proteins, but many of them are variants of green fluorescent proteins (GFPs) derived from Aequorea jellyfish[1–3] or red fluorescent proteins (RFPs) from Discosoma coral[4–6]. While the original coral RFP, DsRed, is obligately tetrameric[7], dimeric and monomeric variants have been created from DsRed[4,8]. RFPs provide the inherent advantages of lower phototoxicity, lower autofluorescence, and deeper tissue penetration associated with longer wavelength excitation light[9]. Therefore, various RFPs are used to visualize live cells in many transgenic RFP-expressing animals such as mice[10,11], rats[12,13], and even larger animals[14,15]. For selective manipulation of specific cells, driver lines expressing Cre recombinase are widely used. The combination of Cre driver mice and Cre-dependent virus vectors is an essential basis for optogenetics and chemogenetics in neuroscience. While recent genome-editing tools such as CRISPR-Cas systems have dramatically changed the generation of transgenic animals[16–18], production and validation of a new Cre driver animal is still time-consuming, especially for large animals. Therefore, it would be useful if we could utilize existing well-characterized RFP-expressing animals to manipulate gene expression in specific cells by using Cre-dependent genetic tools.

Nanobodies are single-chain small antibodies derived from Camelidae such as llamas and they contain only one monomeric antigen-binding unit[19]. Nanobodies and recombinant binders such as designed ankyrin repeat proteins (DARPins)[20,21] are frequently used as small molecular modules that selectively bind to target proteins. Specific recognition of target proteins by nanobodies can be utilized for not only visualization of target proteins[22–24] but also protein degradation[25,26], signal inhibition[27] and property manipulation[28]. Recently, GFP-dependent gene regulation methods using GFP-specific nanobodies have been reported[29,30]. These techniques enable selective gene expression in GFP-expressing cells by utilizing nanobodies as specific binding modules to recognize GFPs. However, these methods have so far been limited to GFP variants and their applications to fluorescent protein-tagged endogenous proteins have not been fully studied.

In this study, we generated Cre recombinase dependent on RFPs (Cre-DOR) using nanobodies and DARPins. We found that functional binding unit pairs for monomeric RFPs (mCherry, mRFP1) are different from those for dimeric RFP (tdTomato). Using AAV vectors, we achieved highly selective expression of genes of interest in vivo. In Est2-mRFP1 mice, we confirmed the utility of Cre-DOR for selective tracing of mRFP1-expressing neurons in mRFP1-expressing transgenic animals. In Grpr-mRFP1 rats, we achieved selective labeling of Grpr-expressing neurons in the medial amygdala and detected a specific neural pathway from the posterodorsal medial amygdala (MePD) to the posterior bed nucleus of the stria terminalis (BSTp). In addition, we found that the activity of Cre-DOR is affected by cellular localization of target RFPs, and we achieved optical and pharmacological control of Cre-DOR activity by utilizing nuclear translocation of target RFPs. Our results advance genetic manipulation tools using nanobodies and DARPins utilizing existing RFP-expressing transgenic animals even if useful Cre-driver lines do not exist.

## Results

### Screening of efficient mCherry-binding protein (MBP) pairs to design Cre recombinase dependent on RFPs. First, we aimed to

construct Cre recombinase dependent on RFPs based on the reported tool named Cre-DOG[30]. In this system, N-terminal and C-terminal split Cre recombinase fragments are fused with specific nanobodies for target proteins, and target proteins mediate reunion of the split Cre recombinase fragments (Fig. 1a). To identify functional pairs of binding proteins, we selected 6 nanobodies and 2 DARPins previously reported to have a highly specific binding property to a monomeric RFP, mCherry[31,32], and we constructed every combination (8 + 8 = 16 constructs). The codon-optimized DNA sequences were synthesized and were inserted instead of GFP-specific nanobodies in pAAV-EF1α-N-CretrcintG (Addgene ID: 69570) or pAAV-EF1α-C-CreintG (Addgene ID: 69571) using NheI and EcoRI sites. We renamed them as MBPs (mCherry-binding proteins) 1 – 8 in this study. MBPs 1-6 are nanobodies and MBPs 7 and 8 are DARPins. As a consequence, we obtained pAAV-EF1α-N-Cre-MBP(1-8)-WPRE and pAAV-EF1α-C-Cre-MBP(1-8)-WPRE.

Then we performed in vitro luciferase reporter assays to find adequate MBP pairs that could induce reunion to reconstruct an active Cre recombinase. N-Cre-MBP, C-Cre-MBP, FLEX-Nano-Luc, and target RFPs were co-transfected into HEK293 cells by the calcium phosphate method. The FLEX switch consists of paired loxP and lox2272 sequences and enables the expression of a gene of interest only when Cre recombinase is functional[33]. NanoLuc is a small and bright luciferase from the deep-sea shrimp Oplophorus gracilirostris[34]. Recombinase activities were measured as luminescence derived from the bioluminescent reaction catalyzed by NanoLuc luciferase. We tested mCherry, mRFP1, and tdTomato as target proteins. These red fluorescent proteins were all derived from the same wild-type DsRed protein[4]. While mCherry and mRFP1 are monomeric, tdTomato is a tandem dimer of two subunits. We also tested mRuby, which is a monomeric variant of the red fluorescent protein eqFP611 derived from Entacmaea quadricolor[35], as a negative control. As shown in Fig. 1b, c, heat maps of mCherry and mRFP1 showed similar patterns. The pair of N-Cre-MBP6 and C-Cre-MBP1 induced high activity for both mCherry and mRFP1. Twin pairs of the same MBP such as N-Cre-MBP1 and C-Cre-MBP1 or N-Cre-MBP2 and C-Cre-MBP2 showed weak reporter activities, possibly indicating competition for the same binding site by both N-Cre-MBP and C-Cre-MBP. In contrast, the heat map for tdTomato greatly differed from those of mCherry and mRFP1 (Fig. 1d). Unfortunately, we did not find any strong signal when we targeted the tetrameric RFP, DsRed (Fig. 1e). The twin pair of N-Cre-MBP8 and C-Cre-MBP8 showed the highest activity for tdTomato. The heat maps for mRuby and No RFPs (Fig. 1f, g) showed only weak recombinase activities around the maps. According to these heat maps obtained from luciferase assays, we selected the pair of N-Cre-MBP6 and C-Cre-MBP1 as a candidate pair for Cre-dependent on monomeric RFP.

### Characterization of recombinase activity of Cre-DORs in vitro.

Next, we investigated the recombination efficiency of the pair of N-Cre-MBP6 and C-Cre-MBP1 (Cre-DOR$^{N6C1}$) using a fluorescent protein reporter. Four kinds of plasmids including N-Cre-MBP6, C-Cre-MBP1, target RFPs, and FLEX-H2B-GFP were co-transfected into HEK293 cells by the calcium phosphate method (Fig. 2a). H2B-GFP shows nuclear localization because H2B (histone 2B) protein binds to the DNA in the nucleus. Recombinase activities were measured as H2B-GFP expression induced by FLEX switching (Fig. 2b). The fluorescent signal of GFP was enhanced by immunostaining using a GFP antibody. Quantitative cell counting of fluorescent images showed that $81.8 \pm 1.5\%$ of mCherry-positive cells were GFP-positive and that $74.1 \pm 1.6\%$ of mRFP1-positive cells were GFP-positive, while $5.6 \pm 0.6\%$ of

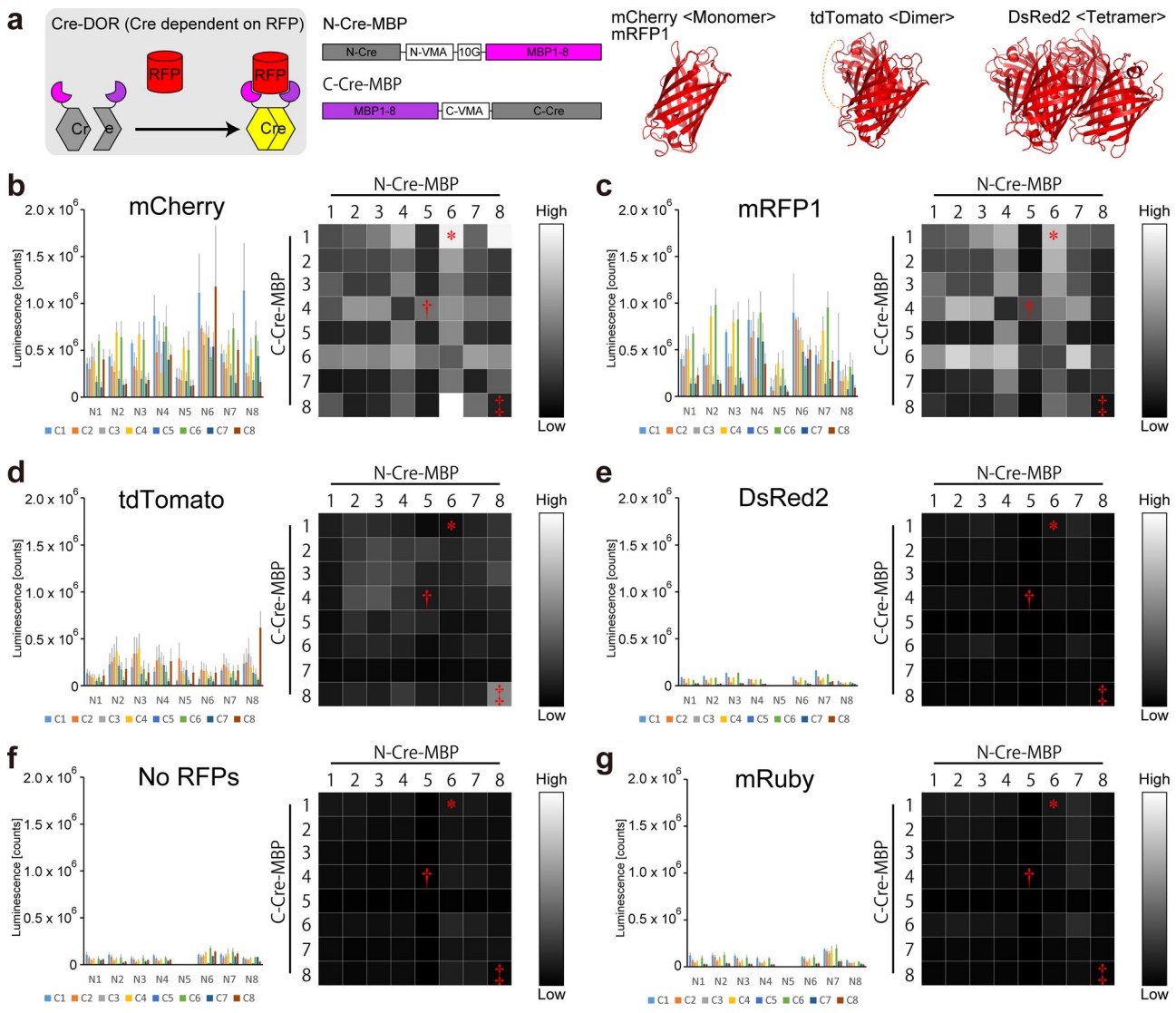

**Fig. 1 Luciferase assay screening of efficient MBP pairs for Cre-DOR. a** Schematic presentation of the construct of Cre-DOR. N- or C-terminal split Cre recombinase is fused with mCherry-binding nanobody or DARPin. Binding of RFPs induces reunion of N-Cre and C-Cre into functional Cre recombinase. While mCherry and mRFP1 are monomeric, tdTomato is dimeric. **b-g** Luciferase assay screening of functional pairs of MBPs for mCherry (**b**), mRFP1 (**c**), tdTomato (**d**), DsRed2 (**e**), No RFPs (**f**) and mRuby (**g**). Every intensity value was normalized by the maximum value and the normalized value is indicated as brightness of 256 levels of gray. VMA: vacuolar membrane ATPase subunit, N-VMA: N-terminal portion of VMA, C-VMA: C-terminal portion of VMA, 10 G: 10 glycine linker. Cre-DOR$^{N6C1}$, Cre-DOR$^{N5C4}$, and Cre-DOR$^{N8C8}$ are marked using *, †, and ‡, respectively.

mRuby-positive cells were GFP-positive ($n = 8$ each) (Fig. 2c, d). The recombination efficiency of Cre-DOR$^{N6C1}$ + mCherry was 14.6-times higher than that of Cre-DOR$^{N6C1}$ + mRuby. The Cre-DOR$^{N6C1}$ system was found to be dependent on all three components of target RFPs, N-Cre-MBP6, and C-Cre-MBP1 in the system. Removal of N-Cre-MBP6 or C-Cre-MBP1 resulted in total loss of reporter activity. Cell counting showed that $0.9 \pm 0.1\%$ of mRFP1-positive cells were GFP-positive without Ccre-MBP1 and that $0.8 \pm 0.1\%$ of mRFP1-positive cells were GFP-positive without Ncre-MBP8 ($n = 8$ each). In all cases, the percentages of RFP-positive cells in GFP-positive cells were higher than 90% possibly because of the transfection method (mCherry: $98.9 \pm 0.3\%$, mRFP1: $98.3 \pm 0.3\%$, mRuby: $91.0 \pm 1.9\%$, mRFP1ΔCCre: $94.4 \pm 3.7\%$, mRFP1ΔNCre: $88.5 \pm 5.6\%$). The heatmaps in Fig. 1 are reliable to predict the efficiency of recombination in HEK293 cells to some extent. For example, we found that Cre-DOR$^{N5C4}$ recognizes mCherry, but much less

effectively mRFP1 in Fig. 2f, and this finding is predictable by heat maps shown in Fig. 1.

We also tested the recombination efficiency of the pair of N-Cre-MBP5 and C-Cre-MBP4 (Cre-DOR$^{N5C4}$). Quantitative cell counting of fluorescent images showed that $57.9 \pm 1.6\%$ of mCherry-positive cells were GFP-positive and that $9.3 \pm 1.1\%$ of mRFP1-positive cells were GFP-positive, while only $0.9 \pm 0.1\%$ of mRuby-positive cells were GFP-positive ($n = 8$ each) (Fig. 2e, f). The recombination efficiency of Cre-DOR$^{N5C4}$ + mCherry was 61.0 times-higher than that of Cre-DOR$^{N5C4}$ + mRuby. The Cre-DOR$^{N5C4}$ system was also found to be dependent on all three components of target RFPs, N-Cre-MBP5, and C-Cre-MBP4 in the system. Removal of N-Cre-MBP5 or C-Cre-MBP4 resulted in total loss of reporter activity. Cell counting showed that $0.7 \pm 0.1\%$ of mRFP1-positive cells were GFP-positive without Ccre-MBP4 and that $0.8 \pm 0.2\%$ of mRFP1-positive cells were GFP-positive without Ncre-MBP5 ($n = 8$ each). Although Cre-

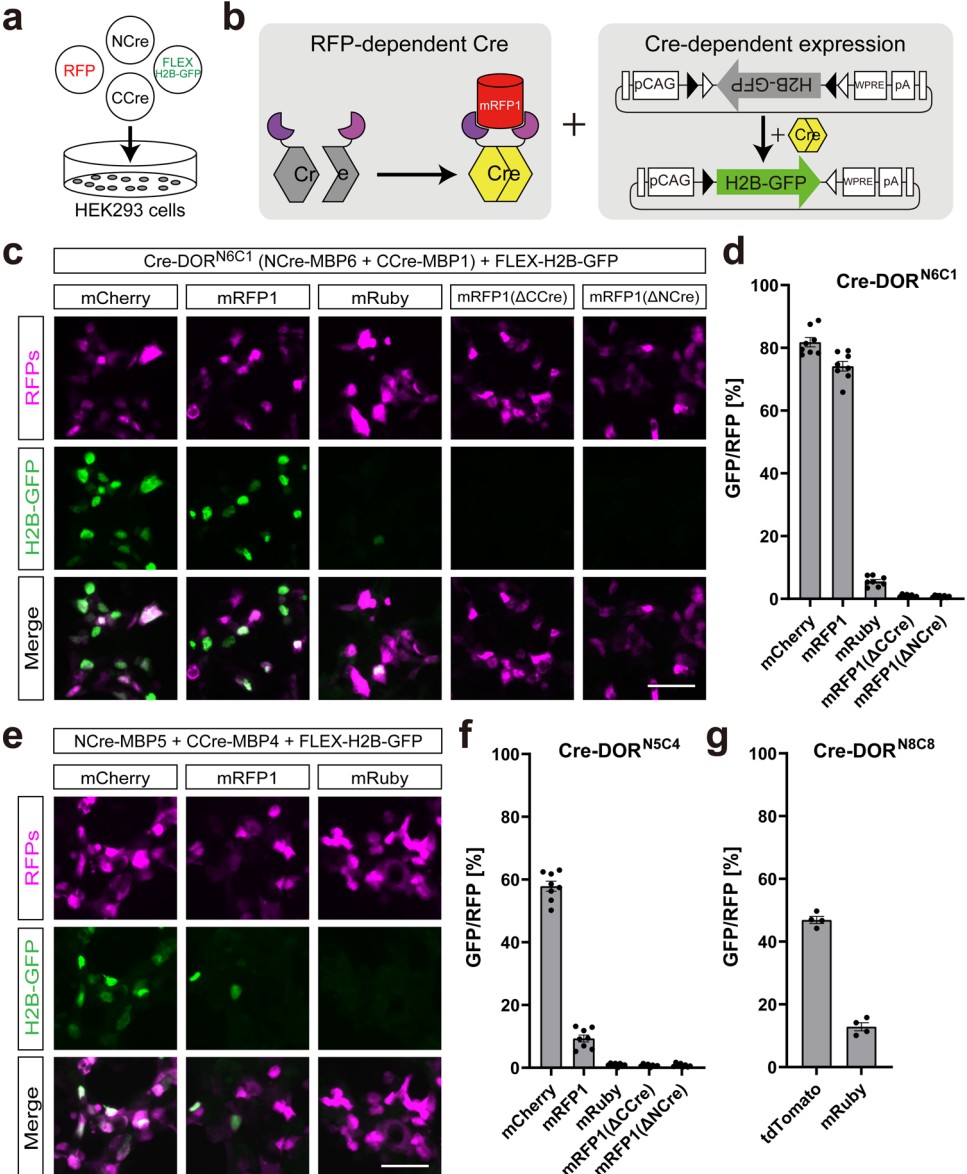

**Fig. 2 Functional assay of Cre-DOR in HEK293 cells. a** Schematic illustration of Cre-DOR transfection in HEK293 cells. Four kinds of plasmids (NCre-MBP, CCre-MBP, FLEX-H2B-GFP, and target RFPs) were transfected in HEK293 cells to assess Cre-DOR recombination efficiency. **b** Illustration of specific expression of H2B-GFP induced by Cre-DOR activated by mRFP1. **c** Fluorescent images of reporter H2B-GFP expression in transfected HEK293 cells to assess Cre-DOR (N-Cre-MBP6 and C-Cre-MBP1) efficiency for target RFPs. **d** Quantification of cell counts in transfected HEK293 cells to assess Cre-DOR (N-Cre-MBP6 and C-Cre-MBP1) efficiency for all components of the system. **e** Fluorescent images of reporter H2B-GFP expression in transfected HEK293 cells to assess Cre-DOR (N-Cre-MBP5 and C-Cre-MBP4) efficiency for target RFPs. **f, g** Quantification of cell counts in transfected HEK293 cells to assess Cre-DOR (N-Cre-MBP5 and C-Cre-MBP4) efficiency for all components of the system. Data are means ± SEM ($n = 8$ or $n = 4$ in **g**). Scale bar = 50 μm.

DOR$^{N5C4}$ can induce more specific recombination dependent on mCherry, we decided to use Cre-DOR$^{N6C1}$ in the later in vivo experiments because of its high efficiency when it was applied for mRFP1.

In the heatmap of the luciferase assay shown in Fig. 1d, we found that the pair of N-Cre-MBP8 and C-Cre-MBP8 shows the highest recombination efficiency dependent on tdTomato. Therefore, we also tested the recombination efficiency of the pair of N-Cre-MBP8 and C-Cre-MBP8 (Cre-DOR$^{N8C8}$). Quantitative cell counting of fluorescent images showed that $46.9 \pm 1.1\%$ of tdTomato-positive cells were GFP-positive, while $12.8 \pm 1.2\%$ of mRuby-positive cells were GFP-positive ($n = 4$ each) (Fig. 2g). Although Cre-DOR$^{N8C8}$ shows mild specificity against tdTomato,

we thought that the S/N ratio of Cre-DOR$^{N8C8}$ was not sufficiently high for use in vivo.

For benchmarking Cre-DOR efficiency, we performed a direct comparison of Cre-DORs with Cre-DOG (Supplementary Fig. 1). Quantitative cell counting of fluorescent images showed that $81.3 \pm 0.8\%$, $72.3 \pm 3.0\%$, and $60.4 \pm 0.8\%$ of the target fluorescent protein-positive cells were reporter fluorescent protein-positive in Cre-DOR$^{N6C1}$, Cre-DOG, and Cre-DOR$^{N5C4}$ experiments, respectively ($n = 4$ each). On the other hand, $6.5 \pm 0.6\%$, $3.9 \pm 0.6\%$, and $0.4 \pm 0.1\%$ of the non-target fluorescent protein-positive cells were reporter fluorescent protein-positive in Cre-DOR$^{N6C1}$, Cre-DOG, and Cre-DOR$^{N5C4}$ experiments, respectively ($n = 4$ each). These results show that Cre-DOR$^{N6C1}$

is more effective than Cre-DOG, while Cre-DOR[N6C1] has more background than Cre-DOG. These results also showed that Cre-DOR[N5C4] is more selective than Cre-DOG, while Cre-DOR[N5C4] is less effective than Cre-DOG.

To check the efficiency of Cre-DOR in the case of a moderate expression rate of target RFPs, we performed similar experiments using an mCherry-expressing HEK293 stable line (Supplementary Fig. 2). This stable line expresses mCherry in more than 90% of the cells (Supplementary Fig. 2a). Four kinds of plasmids including N-Cre-MBP6, C-Cre-MBP1, membrane-bound ALFA tag, and FLEX-nlsGFP were co-transfected into HEK293 cells (Supplementary Fig. 2b). Membrane-bound ALFA tag was transfected to confirm the transfected cells by immunocyto-chemistry using anti-ALFA tag antibody. Quantitative cell counting of fluorescent images showed that $79.8 \pm 3.3\%$ of the ALFA-positive cells were GFP-positive in the mCherry-expressing stable cell line and that $7.5 \pm 2.1\%$ of the ALFA-positive cells were GFP-positive in the control cell line ($n = 4$ each) (Supplementary Fig. 2c, d). These findings are very similar to the results shown in Fig. 2c, d.

It is well known that Cre recombinase and Flp recombinase have orthogonality, and Flp recombinase dependent on GFP (Flp-DOG) has already been reported[36]. Therefore, we performed experiments to investigate the orthogonality of Cre-DOR and Flp-DOG using a mixture of RFP-positive and GFP-positive cells (Supplementary Fig. 3). We confirmed that Cre-DOR induced mRFP1-dependent expression of FLEX-H2B-BFP, while Flp-DOG induced GFP-dependent expression of dFRT-BFP. These results suggest that it can be useful to employ Cre-DOR with GFP-dependent Flp recombinase at the same time to manipulate specific cells expressing GFP and/or RFP in the same animal.

**Cellular localization of target RFPs affects Cre-DOR activity.** Cre recombinase exerts its activity within the nucleus. Therefore, it is possible that cellular localization of target RFPs affects Cre-DOR recombinase activity. If the Cre-DOR system is dependent on nuclear localization of target RFP proteins, it suggests a limitation of Cre-DOR application for selective gene expression in cells expressing membrane proteins fused with RFPs such as channelrhodopsin 2-mCherry or hM3Dq-mCherry in transgenic animals. On the other hand, it also suggests that the Cre-DOR system can be used for clarification of nuclear translocation of proteins such as nuclear receptors.

To test this hypothesis, we created mCherry and tdTomato fused with various localization signal peptides: CAAX motif for membrane localization, NES for cytosolic localization, and NLS for nuclear localization (Fig. 3a). We observed clear intracellular translocalization of mRFP1 and tdTomato by being fused with these motifs. mCherry/tdTomato fused with CAAX were localized in the plasma membrane, mCherry/tdTomato fused with NES were localized in the cytosol, mCherry/tdTomato without any motif were localized in both the cytosol and nucleus, and nls-mCherry/tdTomato were localized in the nucleus (Fig. 3b). Next, we performed in vitro luciferase reporter assays to investigate the effect of cellular translocation on recombinase activity. Plasmids for N-Cre-MBP, C-Cre-MBP, FLEX-NanoLuc, and target RFPs were co-transfected into HEK293 cells. When Cre-DOR[N6C1] or Cre-DOR[N8C8] targets RFPs with localization signals, the luciferase assay showed a clear difference between the cellular localizations (Fig. 3c). A gradual incremental tendency of recombination activity among membrane-bound, cytosolic, and nucleic localization of target RFPs was observed. Quantitative analyses of luciferase assay data showed that recombinase activity is significantly different between Cre-DOR[N6C1]/Cre-DOR[N8C8] + mCherry/tdTomato-NES and Cre-DOR[N6C1]/Cre-DOR[N8C8] + nls-mCherry/tdTomato ($P < 0.0001$, Tukey's

multiple comparison test, $n = 11$). Note that average fluorescence intensity of mCherry-NES or tdTomato-NES is higher than that of nls-mCherry or nls-tdTomato.

Next, we aimed to control Cre-DOR activity by using chemical ligands. Glucocorticoid receptor (GR) is a nuclear receptor and it is translocated into the nucleus after binding its ligand, glucocorticoid[37] (Fig. 3d). Ligand-induced translocation of GR has been detected by addition of a GFP to the N-terminus of GR[38,39]. Therefore, we attached RFPs to human glucocorticoid receptor alpha (hGRα) to control Cre-DOR activity. We found that mCherry-GR was localized in the cytosol without its ligand, dexamethasone (Dex) and that it translocated to the nucleus after incubation with Dex (1 µM, 1 h) (Fig. 3e). In Fig. 3e, nuclei are visualized by fluorescence from Histone 2B-BFP (pAAV-EF1α-H2B-BFP-WPRE) that was co-transfected with mCherry-GR. Cre recombinase activity of Cre-DOR[N6C1] targeting mCherry-GR was strongly increased by the presence of Dex (Fig. 3f).

An increase in Cre recombinase activity by nuclear localization of RFPs suggests that Cre-DOR activity can be manipulated by controlling the intracellular localization of target RFPs. To test this hypothesis, we used a light-inducible nuclear export domain called LEXY[40]. LEXY consists of an engineered LOV2 domain from *Avena sativa* phototropin-1 (AsLOV2), in which the C-terminal Jα helix was converted into an artificial NES. In the dark, the NES is tightly packed against the AsLOV2 core and is thus inactive. Exposure to blue light induces unfolding of the modified Jα helix, uncovering the NES (Fig. 3g). We confirmed that nls-mCherry-LEXY was localized mainly in the nucleus in the dark, and blue light illumination (465 nm, 7 W/m$^2$) induced translocation of fused RFPs into the cytosol in 30 min (Fig. 3h). In Fig. 3h, nuclei are visualized by fluorescence from Histone 2B-BFP that was co-transfected with nls-mCherry-LEXY. We found that blue light illumination inhibited Cre recombination activity of Cre-DOR[N6C1] targeting nls-mCherry-LEXY as indexed by luciferase activity (Fig. 3i).

**Functional assay of Cre-DOR in vivo using AAV vectors.** To examine whether Cre-DOR[N6C1] functions in living animals, we generated AAV vectors encoding N-Cre-MBP6, C-Cre-MBP1, FLEX-nlsGFP, and target RFPs. GFP tagged with a nuclear localization signal (nlsGFP) is localized mainly in the nucleus. 600 nl of a mixture of virus vectors for Cre-DOR[N6C1], FLEX-nlsGFP ($1 \times 10^{12}$ vg/ml each), and target mRFP1 ($5 \times 10^{10}$ vg/ml) was unilaterally injected into the right side M1 cortex of wild-type 10-week-old male mice (Fig. 4a). The titer of the target RFP-expressing vector was lowered to induce scattered expression and make it easy to count fluorescent protein-expressing cells separately. Recombinase activities were measured as nlsGFP expression induced by FLEX switching (Fig. 4b). Four weeks after injection, mice were sacrificed for immunohistochemistry and brain slices were stained with anti-GFP. Clear expression of nlsGFP at the injected site in the M1 cortex was observed (Fig. 4c). Quantitative cell counting of fluorescent images showed that $50.2 \pm 2.5\%$ of mRFP1-positive cells were GFP-positive and that $93.5 \pm 0.6\%$ of GFP-positive cells were mRFP1-positive ($n = 5$ each) (Fig. 4d, e). The expression efficiency of GFP in the center area of injection was higher than that in the peripheral area. To check the specificity of the Cre-DOR[N6C1] system in vivo, we injected vectors for Cre-DOR[N6C1], FLEX-nlsGFP ($1 \times 10^{12}$ vg/ml each), and mRuby ($5 \times 10^{10}$ vg/ml) as a control (Fig. 4f). While we observed comparative amounts of mRuby-expressing neurons in the M1 cortex, we found only sparce nlsGFP-expressing neurons in the same area (Fig. 4g). Quantitative cell counting of nlsGFP and mRuby-positive cells showed a clear difference between Cre-DOR[N6C1] + mRFP1-injected mice and

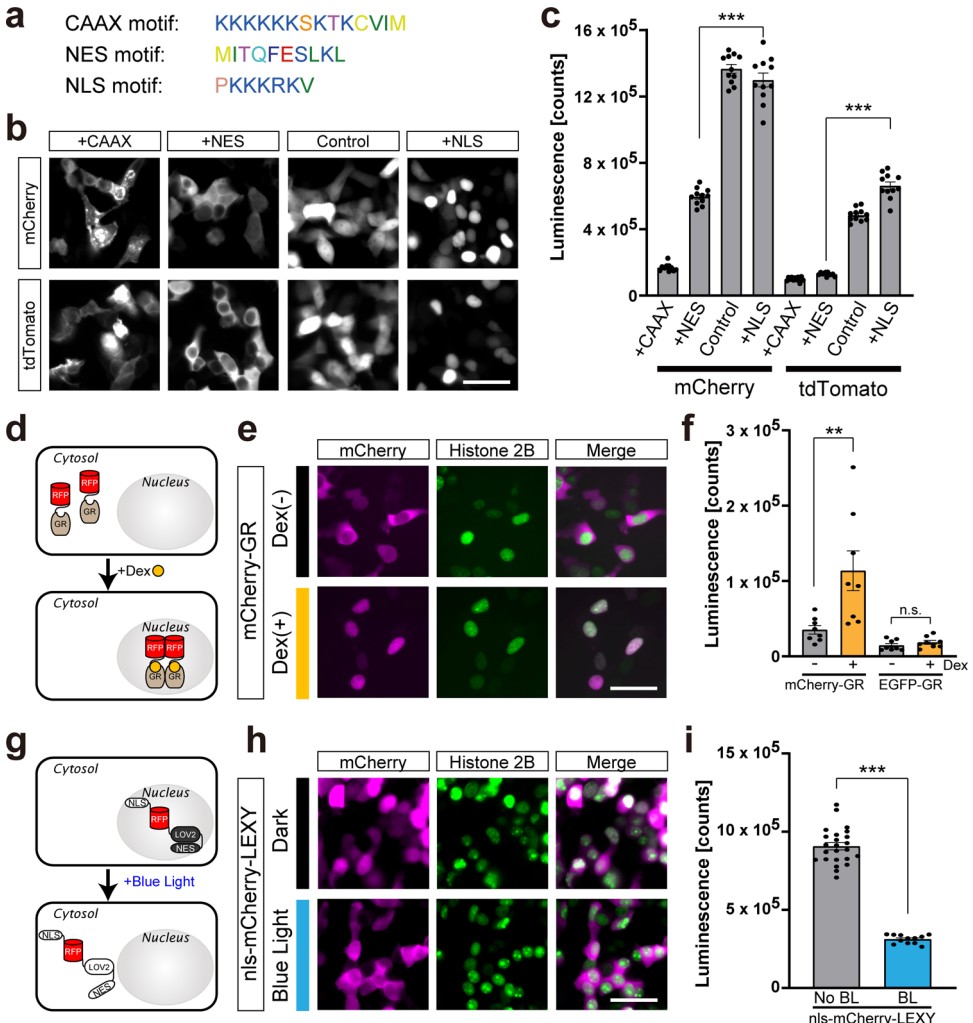

**Fig. 3 Cellular localization and recombinase activity of Cre-DOR. a** Amino acid sequences of the CAAX motif (membrane localization signal), NES motif (nuclear export signal), and NLS motif (nuclear localization signal). **b** Fluorescent images of mCherry or tdTomato with or without (Control) each localization motif. Scale bar = 50 μm. **c** Luciferase assay of Cre-DOR^{N6C1} or Cre-DOR^{N8C8} with RFPs having different cellular localizations. Data are means ± SEM. Statistical analyses were performed by one-way ANOVA followed by Tukey's multiple comparison test ($n = 11$, ***$P < 0.001$). **d** Schematic representation of ligand-induced translocation of RFPs from the cytosol to the nucleus. Dexamethasone (Dex) induces translocation of GRs upon its binding. **e** Fluorescent images of mCherry-GR without or with Dex (1 μM, 1 h). Scale bar = 50 μm. **f** Luciferase assay of Cre-DOR^{N6C1} activity targeting mCherry-GR without or with Dex (1 μM, 24 h). Data are means ± SEM. Statistical analyses were performed by one-way ANOVA followed by Tukey's multiple comparison test ($n = 8$, **$P < 0.01$). **g** Schematic representation of light-induced translocation of RFPs from the nucleus to the cytosol. The LEXY domain consists of a modified AsLOV2 domain with NES. Blue light induces exposure of the NES and results in translocation of RFPs. **h** Fluorescent images of nls-mCherry-LEXY with or without blue light illumination (465 nm, 3.7 W/m²). Scale bar = 50 μm. **i** Luciferase assay of Cre-DOR^{N6C1} activity targeting nls-mCherry-LEXY without or with blue light illumination ($n = 24, 12$). Data are means ± SEM. Statistical analyses were performed by Student's *t*-test (***$P < 0.001$). In Fig. 3e, h, nuclei are visualized by fluorescence from H2B-BFP that was co-transfected with mCherry-GR or nls-mCherry-LEXY.

Cre-DOR^{N6C1} + mRuby-injected mice. The cell counting showed that 1.5 ± 0.5% of mRuby-positive cells were GFP-positive and that 13.3 ± 4.1% of GFP-positive cells were mRuby-positive ($n = 5$ each) (Fig. 4d, e). The recombination efficiency of Cre-DOR^{N6C1} + mRFP1 was 34.4-times higher than that of Cre-DOR^{N6C1} + mRuby in these experiments. All these data suggest in vivo specificity of the Cre-DOR^{N6C1} system using AAV vectors.

We also confirmed that Cre-DOR can be used in combination with intravenous injection of AAV vectors. AAV-EF1a-mRFP1-WPRE or AAV-EF1a-mRuby-WPRE (200 ul; $2 \times 10^{11}$ vg/mouse) was injected systemically (Supplementary Fig. 4a). They are packaged by AAV PHPeb and can infect through the blood-brain barrier. Within the same day, 600 nl of a mixture of virus vectors for Cre-DOR^{N6C1} and FLEX-nlsGFP ($1 \times 10^{12}$ vg/ml each) was

unilaterally injected into the right side M1 cortex of wild-type 10-week-old male mice (Supplementary Fig. 4a). Four weeks after injection, mice were sacrificed for immunohistochemistry and brain slices were stained with anti-GFP. Clear expression of nlsGFP at the injected site in the M1 cortex was observed (Supplementary Fig. 4b). While we observed comparative amounts of mRuby-expressing neurons in the M1 cortex, we found only sparce nlsGFP-expressing neurons in the same area. Quantitative cell counting of fluorescent images showed that 49.9 ± 3.9% of the mRFP1-positive cells were GFP-positive and that 93.3 ± 0.4% of the GFP-positive cells were mRFP1-positive ($n = 4$ each) (Supplementary Fig. 4c). Quantitative cell counting of nlsGFP and mRuby-positive cells showed a clear difference between Cre-DOR^{N6C1} + mRFP1-injected mice and Cre-DOR^{N6C1} + mRuby-injected mice. The cell counting showed

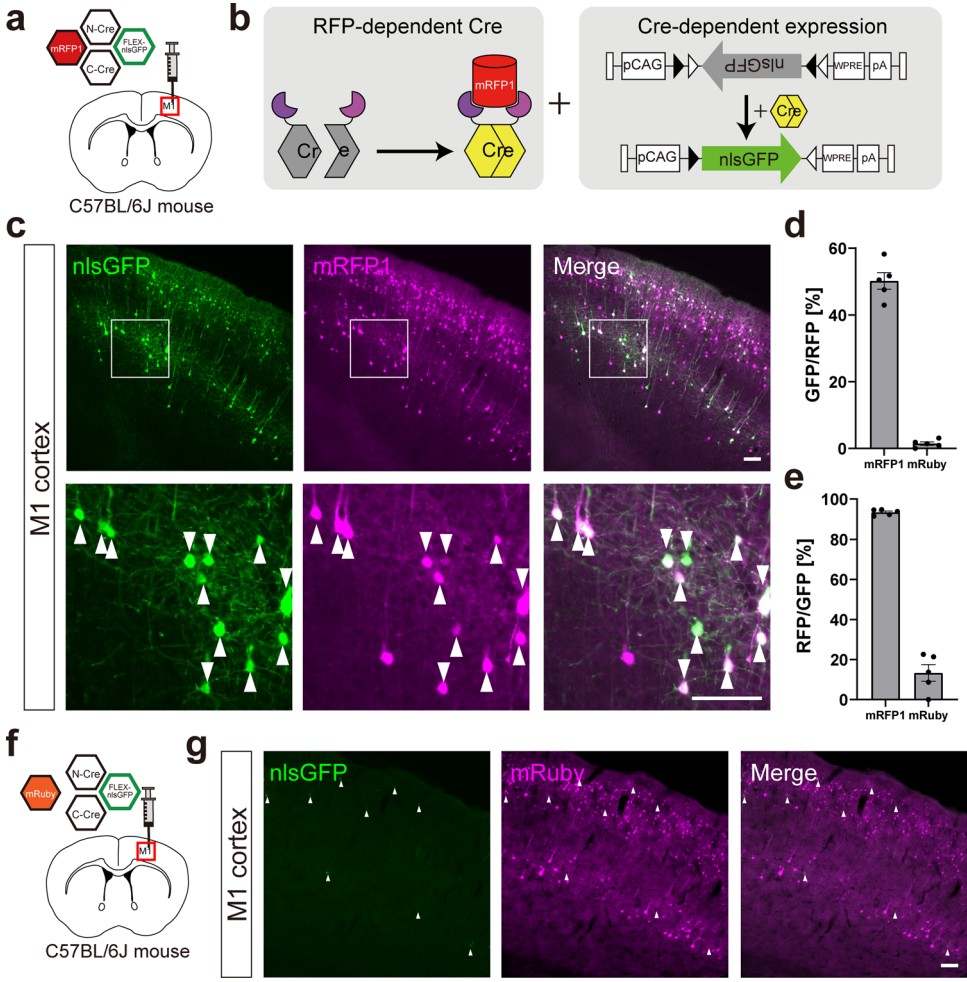

**Fig. 4 Functional assay of Cre-DOR$^{N6C1}$ in vivo. a** Injection schema of the Cre-DOR$^{N6C1}$ test with mRFP1 in wild-type mice. Four kinds of virus (NCre-MBP6, CCre-MBP1, FLEX-nlsGFP, and mRFP1) were injected in the M1 cortex at the same time. **b** Schematic representation of specific expression of nlsGFP induced by Cre-DOR activated by mRFP1. **c** Fluorescent images of the M1 cortex. Scale bar = 100 μm. **d** Quantification of cell counts to assess the efficiency of Cre-DOR$^{N6C1}$ (n = 5 each). **e** Quantification of cell counts to assess the fidelity of Cre-DOR$^{N6C1}$ (n = 5 each). **f** Injection schema of the Cre-DOR$^{N6C1}$ test with mRuby in wild-type mice. **g** Fluorescent images of the M1 cortex in which the four viruses were injected. Scale bar = 100 μm. Data are means ± SEM.

that 1.7 ± 0.7% of the mRuby-positive cells were GFP-positive and that 10.5 ± 3.6% of the GFP-positive cells were mRuby-positive (n = 4 each) (Supplementary Fig. 4d). The recombination efficiency of Cre-DOR$^{N6C1}$ + mRFP1 was 29.9-times higher than that of Cre-DOR$^{N6C1}$ + mRuby in these experiments. All of these data confirmed in vivo specificity of the Cre-DOR$^{N6C1}$ system using AAV vectors.

**Functional assay of Cre-DOR in mRFP1-expressing transgenic mice.** Next, we examined selective expression by Cre-DOR$^{N6C1}$ in mRFP1-expressing transgenic animals. In *Esr2*-mRFP1 mice, neurons in the paraventricular nucleus (PVN) are visualized by mRFP1. 1 μl of a mixture of AAV9-EF1α-NCre-MBP6-WPRE (6 × 10$^{12}$ vg/ml), AAV9-EF1α-CCre-MBP1-WPRE (6 × 10$^{12}$ vg/ml) and AAV9-CAG-FLEX-palGFP-WPRE (6 × 10$^{12}$ vg/ml) was injected in the PVN of *Esr2*-mRFP1 transgenic mice (Fig. 5a). Recombinase activities were measured as expression of GFP tagged with a palmitoylation signal (palGFP) induced by FLEX switching (Fig. 5b). palGFP is sorted to the plasma membrane and has been used to trace neuronal fibers anterogradely[41,42]. Four weeks after injection, mice were sacrificed for immunohistochemistry and brain slices were stained with anti-GFP and anti-mRFP1. Clear expression of palGFP at the injected site in the

PVN was observed (Fig. 5c). It has been reported that mRFP1-expressing neurons in the PVN of *Esr2*-mRFP1 mice include oxytocin neurons[43] and oxytocin neurons send their axons into the posterior pituitary. In accordance with these previous findings, we observed clear projection from the palGFP-expressing neurons in the PVN and dense axonal terminals in the posterior pituitary (Fig. 5d). Quantitative cell counting of palGFP-positive cells in the PVN showed a clear difference between mRFP(+) mice and mRFP(−) mice (Fig. 5e). The cell counting showed that 24.1% of mRFP1-positive neurons in the PVN express palGFP on average (Fig. 5f). These results showed the usability of Cre-DOR$^{N6C1}$ for detection of selective neural projection in mRFP1-expressing transgenic animals. We also confirmed that Cre-DOR$^{N6C1}$ can be functional in other parts of the brain such as the islands of Calleja (ICj) in *Esr2*-mRFP1 mice (Supplementary Fig. 5).

**Anterograde tracing of mRFP1-expressing neurons in *Grpr*-mRFP1 rats.** Finally, we examined selective expression by Cre-DOR$^{N6C1}$ in mRFP1-expressing transgenic animals. In gastrin-releasing peptide receptor (*Grpr*)-mRFP1 transgenic rats, neurons in the posterior amygdala are visualized by mRFP1. 1 μl of a mixture of AAV9-EF1α-NCre-MBP6-WPRE (6 × 10$^{12}$ vg/

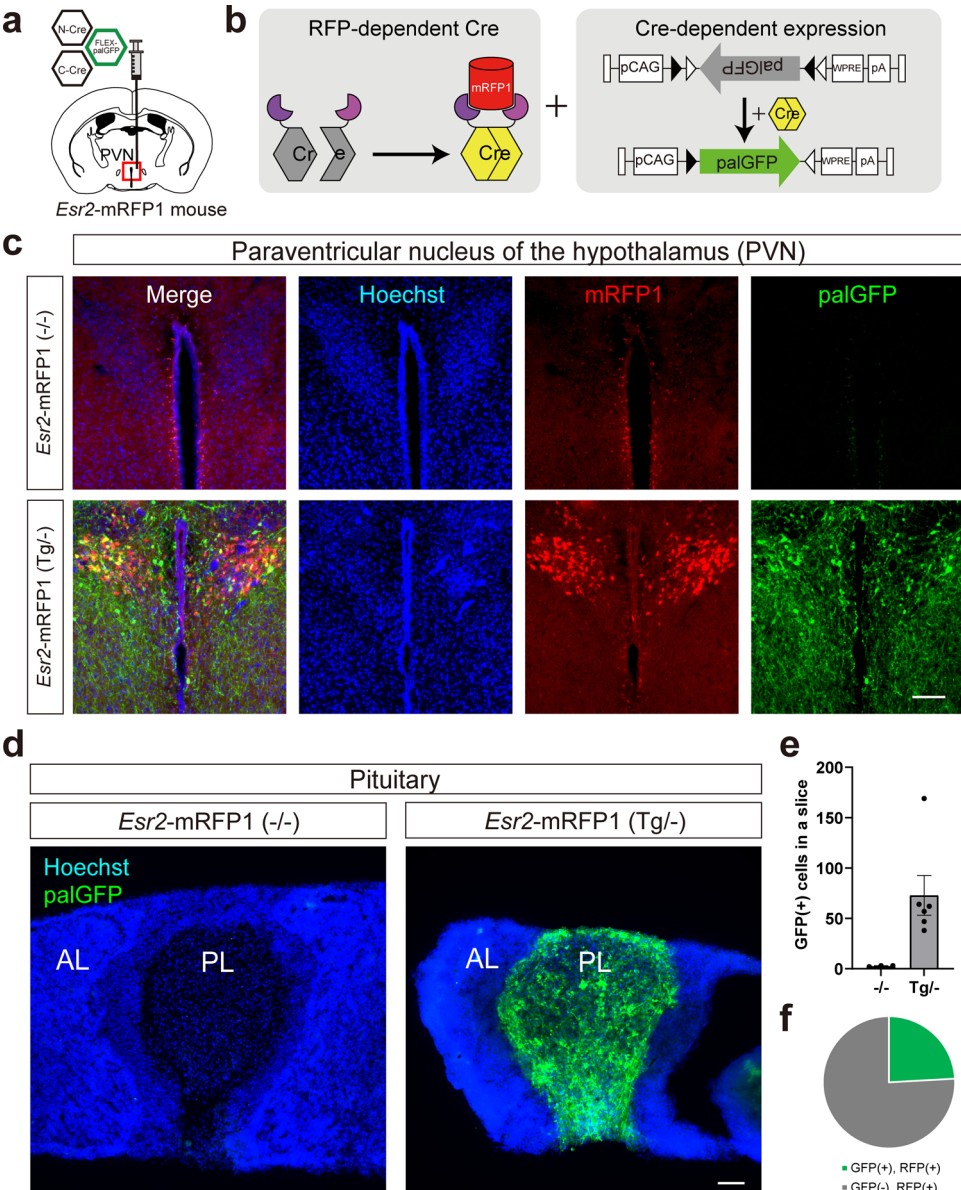

**Fig. 5 Anterograde tracing using Cre-DOR in *Esr2*-mRFP1 transgenic mice. a** Stereotaxic injection schema of Cre-DOR^N6C1 in *Esr2*-mRFP1 transgenic mice. Three kinds of virus (NCre-MBP6, CCre-MBP1, FLEX-palGFP) were injected in the paraventricular nucleus (PVN). **b** Schematic representation of specific expression of palGFP induced by Cre-DOR activated by mRFP1. **c** Immunofluorescent images of the PVN of injected mice. Scale bar = 100 μm. **d** Representative images of axonal projection from palGFP-expressing neurons in the PVN of *Esr2*-mRFP1 transgenic mice. AL anterior lobe, PL posterior lobe. Scale bar = 100 μm. **e** Quantification of mRFP1-induced expression of palGFP in WT and Tg *Esr2*-mRFP1 *mice*. **f** Cell counting results showing the average percentage of palGFP-positive cells in the PVN of the Cre-DOR virus-injected *Esr2*-mRFP1 transgenic mice.

ml), AAV9-EF1α-CCre-MBP1-WPRE (6 × 10^12 vg/ml) and AAV9-CAG-FLEX-palGFP-WPRE (6 × 10^12 vg/ml) was injected in the medial amygdala area of male *Grpr*-mRFP1 transgenic rats (Fig. 6a). Four weeks after injection, rats were sacrificed for immunohistochemistry and brain slices were stained with anti-GFP and anti-mRFP1. Clear and selective expression of palGFP at the injected site in the posterodorsal medial amygdala (MePD) was observed (Fig. 6b). We observed clear projection from the palGFP-expressing neurons in the MePD. We found a bundle of smooth passing fibers in the stria terminalis (ste) and dense axonal terminals with varicosity in the posterior bed nucleus of the stria terminalis (BSTp or STMP) (Fig. 6c). These findings suggest that mRFP-expressing neurons in the MePD send their axons to the BSTp (Fig. 6e). Finally, we confirmed this neural projection using a retrograde tracer. We injected 300 nl of green

retrobeads in the BSTp of *Grpr*-mRFP1 transgenic rats (Fig. 6d). One week after injection, rats were sacrificed. We detected some mRFP1 neurons that included green retrobeads in the MePD. These results support our idea that mRFP-expressing neurons in the MePD send their axons to the BSTp (Fig. 6f).

To visualize each neuronal morphology, we also performed sparse labeling of mRFP-expressing neurons in the preoptic area of *Grpr*-mRFP1 transgenic rats. Three μl of a mixture of AAV9-EF1α-NCre-MBP6-WPRE (3 × 10^11 vg/ml), AAV9-EF1α-CCre-MBP1-WPRE (3 × 10^11 vg/ml) and AAV9-CAG-FLEX-hrGFP-WPRE (3 × 10^11 vg/ml) was unilaterally injected into the preoptic area of 10–15–week-old male *Grpr*-mRFP1 transgenic rats (Supplementary Fig. 6a). Although the expression of hrGFP was sparse, 92.5% of the hrGFP-expressing neurons were confirmed to be mRFP1-expressing neurons. It is likely that Grpr-positive

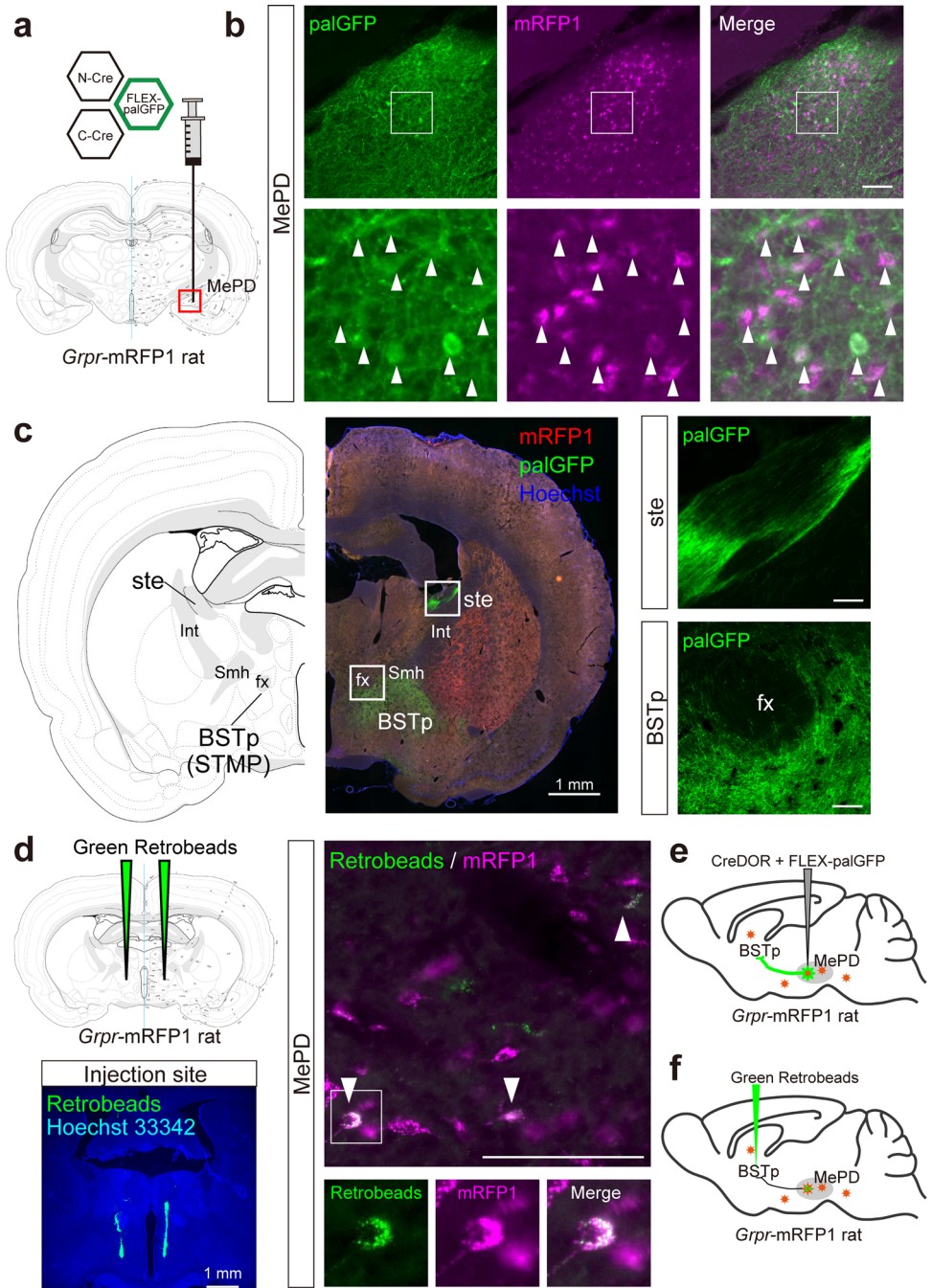

**Fig. 6 Anterograde tracing using Cre-DOR in transgenic mRFP1-expressing rats. a** Stereotaxic injection schema of Cre-DOR[N6C1] in *Grpr*-mRFP1 transgenic rats. Three kinds of virus (NCre-MBP6, CCre-MBP1, FLEX-palGFP) were injected in the medial amygdala. **b** Immunofluorescent images of the MePD of injected rats. Scale bar = 100 μm. **c** A representative image of axonal projection from palGFP-expressing neurons in the MePD of a *Grpr*-mRFP1 transgenic rat. Scale bar = 100 μm or 1 mm. **d** Injection schema of green retrobeads in a *Grpr*-mRFP1 rat. Scale bar = 100 μm or 1 mm. **e** Illustration of anterograde tracing using Cre-DOR in *Grpr*-mRFP1 rats. **f** Illustration of retrograde tracing using green retrobeads in *Grpr*-mRFP1 rats. All the rat brain atlas images were derived from Swanson, L.W. (2004) Brain maps: structure of the rat brain, 3rd edition (Creative Commons Attribution-NonCommercial 4.0 International License: https://creativecommons.org/licenses/by-nc/4.0/) with permission of Dr. Swanson.

neurons in the preoptic area are peptidergic neurons from their neuroanatomical features. Numerous axonal varicosities and aspiny dendrites are features of many peptidergic neurons. Indeed, hrGFP-labeled neurons appear to be aspiny (Supplementary Fig. 6b), and we found numerous axonal varicosities labeled with hrGFP in the preoptic area (Supplementary Fig. 6b, arrowheads) or more caudal and ventrolateral region from the injection site of the preoptic area.

## Discussion

In this study, we generated Cre recombinase dependent on RFPs (Cre-DOR) by making use of split-Cre combined with nanobodies and DARPins as binding units for target RFPs (Fig. 1). Cre recombinase activity was selectively induced by a monomeric RFP or by a dimeric RFP in vitro (Fig. 2). The efficiency of Cre-DOR is affected by intracellular localization of target RFPs, and we achieved optical and pharmacological manipulation of Cre-DOR

activity by utilizing intracellular translocation of the light-inducible nuclear export system (LEXY) and glucocorticoid receptor (Fig. 3). Using AAV vectors, we confirmed the efficiency and fidelity of selective induction of Cre-DOR activity in the mouse brain (Fig. 4). Cre-DOR induced mRFP1-dependent expression of palGFP in Esr2-mRFP1 transgenic mice (Fig. 5). We also achieved anterograde tracing of RFP-expressing neurons using Cre-DOR and found neural projection from mRFP-expressing neurons in the MePD to the BSTp in Grpr-mRFP1 transgenic rats (Fig. 6).

Our results provide a method for utilizing existing RFP transgenic animal lines to regulate gene expression selectively in RFP-expressing cells. The number of available transgenic large animals including non-human primates has been gradually increasing[15,44]. This technique is useful not only because it can be used as a substitute of Cre transgenic lines but also because it can fundamentally provide a unique way to mark RFP-expressing cells. Although genome editing and other genetic techniques are being rapidly developed, it is still time-consuming to generate new transgenic animals, especially large animals. It is common to use 2 A peptide for simultaneous expression of both Cre recombinase and fluorescent proteins in transgenic animals[45,46]. However, a tandem sequence linked with 2 A peptide can induce unpredictable cis regulation of expression in some cases[47]. Therefore, this simultaneous expression of Cre recombinase and fluorescent proteins using 2 A peptide is not complete substitute for RFP-transgenic animals. Our method provides an efficient way to use the same transgenic line for different purposes. In vitro experiments using HEK293 cells (Fig. 2 and Supplementary Figs. 1, 2) suggest that approximately 80% of transfected cells can induce RFP-dependent recombination by Cre-DOR$^{N6C1}$, while approximately 8% of transfected cells can induce RFP-independent recombination. Therefore, we recommend using Cre-DOR$^{N5C4}$ when leakage expression should be strictly prevented, while leakage expression in vivo is substantially improved in experiments using AAV vectors (Figs. 4, 5, 6 and Supplementary Figs. 3, 4).

Our findings using Cre-DOR provide valuable information on specific neural pathways in mRFP1-expressing transgenic animals. In Esr2-mRFP1 transgenic mice, we found that estrogen receptor β-expressing neurons in the PVN send their axons in the posterior pituitary (Fig. 5d). We also reported that 70–80% of oxytocin neurons expressed mRFP1, while only 10% of vasopressin neurons did so in the PVN of Esr2-mRFP1 transgenic mice[43]. Estrogen has been found to both reduce anxiety-related behaviors and increase oxytocin peptide transcription, suggesting a role for oxytocin in this estrogen receptor β-mediated anxiolytic effect[48]. Considering that estrogen receptor α is not expressed in oxytocin neurons[49], it is reasonable to assume that estrogen receptor β in oxytocin neurons in the PVN plays a main role in this modulation. The absence of good antibodies for immunostaining for estrogen receptor β and the nuclear localization of estrogen receptor β hinder selective neural tracing of estrogen receptor β-expressing neurons in the PVN so far. Our findings provide a good way to investigate the physiological role of this specific neural pathway.

In Grpr-mRFP1 transgenic rats, we found that Grpr-expressing neurons in the MePD send their axons to the BSTp (Fig. 6c). Grpr neurons play an important role in emotional responses, social interaction, and feeding behavior[50]. Although Grpr neurons in the lateral amygdala are frequently investigated, Grpr neurons in the medial amygdala are not well studied so far. It has been reported that the anterior and posterior medial amygdala differentially innervate downstream targets. While the anterior medial amygdala densely innervates the horizontal diagonal band of Broca and the medial olfactory tubercle, the posterior medial amygdala innervates the BNST[51]. Our findings using Cre-DOR support these previous anatomical works and suggest a functional linkage between Grpr and sexual behaviors.

We generated monomeric/dimeric state-specific RFP-dependent Cre (Fig. 2). Functional properties of MBP pairs seem to reflect the difference of oligomerization states among mCherry, mRFP1, tdTomato, and DsRed. Theoretically, it is reasonable to assume that pairs of identical MBPs do not work for monomeric RFPs because of binding competition on a single recognition site. Indeed, we found low luciferase activity when we used mCherry or mRFP1 as a target protein for pairs of identical MBPs (Fig. 1b, c). However, we found that the combination of N-Cre-MBP8 and C-Cre-MBP8 is a functional pair for tdTomato. These findings show that it can be essential to prepare multiple nanobodies for designing nanobody-dependent molecular tools, while it is possible to make a Cre-DOR-like system using only one nanobody if the target proteins form homodimers within the cell. Since tdTomato is a dimeric protein, the same recognition site can be exposed at two locations in one dimer at the same time. We think that such a method might be useful for simultaneous achievement of fluorescence imaging and optical genetic manipulation, although there are several reported light-activated Cre systems such as PA-Cre[52].

We revealed that the Cre-DOR system is dependent on nuclear localization of target RFP proteins (Fig. 3). Our findings show a limitation of Cre-DOR application. It might be difficult to use our Cre-DOR system for selective gene expression in cells expressing membrane protein fused with RFPs such as channelrhodopsin 2-mCherry or hM3Dq-mCherry. On the other hand, our findings suggest that the Cre-DOR system can be used for clarification of nuclear translocation of proteins such as nuclear receptors. Cre-DORs showed minimal activity for cell membrane-localized target RFPs and high activity for nucleus-localized RFPs (Fig. 3c). Therefore, it might be a good way to target membrane proteins that change their localization from the plasma membrane to the nucleus. In Notch signaling, the intercellular domain of Notch (NICD), which is a single-pass transmembrane receptor protein, is cleaved and translocates into the nucleus[53]. Amyloid beta is derived from amyloid precursor protein (APP) cleavage by γ-Secretase. Another cleavage fragment, APP intracellular domain (AICD), is also known to translocate into the nucleus, inducing the expression of related genes[54]. With specific binding proteins, these molecules can be targeted by a similar method for specific gene manipulation of signal-on cells in the future.

In the future, it will be possible for endogenous proteins to be used as targets for split-Cre systems. Our results showed that not only nanobodies but also DARPins can be used to construct functional molecules utilizing selective binding to target proteins for reunion of split-Cre. Therefore, when multiple specific nanobodies or other binding proteins exist, it might be possible to induce selective gene expression dependent on a target endogenous protein. Indeed, we can easily find multiple promising nanobodies for endogenous proteins in an open website database[55]. In addition, rapid approaches to generate large repertoires of recombinant nanobodies and de novo proteins designed for creating customized small binding proteins have been reported[31,56]. A recent paper also showed that machine learning using only structural information of a target protein can produce specific small-sized specific binding proteins[57]. An endogenous protein can be used as a target protein for reunion of split-luciferase fused with specific nanobodies[58]. GFP-dependent Flp recombinase has also been reported[36]. It will be useful to employ Cre-DOR with GFP-dependent Flp recombinase at the same time to manipulate two kinds of cells in the same animal (Supplementary Fig. 3). This result suggests that it might be possible to perform Boolean operation using Cre-DOR and Flp-DOG. For example, if we use pAAV-hSyn Con/Fon

hChR2(H134R)-EYFP[59], we can manipulate only GFP- and RFP-expressing cell populations. Taken together, our results suggest a potential for AAV vectors for target protein-based genetic manipulation based on AAV vectors.

## Materials and methods

**Animals.** All experimental procedures with mice were approved by the Institutional Animal Experiment Committee of Jichi Medical University. Male C57BL/6 J mice were purchased from Charles River Laboratories Japan (Kanagawa, Japan). The mice were maintained under a 12-h light/dark cycle (light period: 7:30-19:30, dark period: 19:30-7:30) in a room with controlled temperature (22 ± 2 °C) and humidity (55 ± 15%). Food and water were available *ad libitum*. *Esr2*-mRFP1 transgenic mice were generated by Dr. Hirotaka Sakamoto and their validation was published[43]. *Grpr*-mRFP1 transgenic rats were also generated by Dr. Hirotaka Sakamoto and their validation was published[60]. Briefly, the *Grpr* promoter-human heparin-binding epidermal growth factor-like growth factor-2A-mRFP1 BAC transgene was purified for microinjection using a slight modification of the procedure described previously[61]. *Grpr*-mRFP1 transgenic rats were generated by pronuclear injection of Wistar rat embryos (Institute of Immunology Co., Ltd., Tokyo, Japan). For experiments using *Esr2*-mRFP1 transgenic mice and *Grpr*-mRFP1 transgenic rats, adult transgenic animals bred in the animal facility of Okayama University were used. All of the *Grpr*-mRFP1 rats and *Esr2*-mRFP1 mice were maintained on a 12-h light/12-h dark cycle and were provided unlimited access to water and rodent chow. The Committee for Animal Research, Okayama University, Japan authorized the experimental procedures.

**Drug administration and light illumination.** Dexamethasone (Dex) was purchased from Merck (Darmstadt, Germany). Dex was dissolved in ethanol as a 1 mM stock solution and diluted with culture medium to 1 μM just after plasmid transfection. An LED flat panel light (TH2-100X100BL; CCS Inc., Kyoto, Japan) was used for uniform blue light illumination (465 nm, 3.7 W/m²) of LEXY domains. The procedures for blue light stimulation were based on a previous report[62]. In Fig. 3e, h, nuclei are visualized by fluorescence from Histone 2B-BFP (pAAV-EF1α-H2B-BFP-WPRE) that was co-transfected with mCherry-GR or nls-mCherry-LEXY.

**DNA construction.** Amino acid sequences of mCherry nanobodies (MBPs 1–6) were described in a previous report[31] (originally described as LaMs 1–4, 6, and 8). DNA sequences of mCherry DARPins (MBP 7 and MBP 8) were described in another previous report[32] (originally described as 2m22 and 3m160). DNA sequences of MBPs 1–8 were codon-optimized, synthesized (Genscript, Piscataway, NJ), and inserted instead of GFP-specific nanobodies in pAAV-EF1α-N-CretrcintG (Addgene ID: 69570) or pAAV-EF1α-C-CreintG (Addgene ID: 69571) using NheI and EcoRI sites. Amino acid sequences of cellular localization signals were derived from previous reports (CAAX motif[52], NES[63], and NLS[64]). GFP tagged with a palmitoylation signal (palGFP) has been used to efficiently trace neuronal fibers[41]. A vector coding palGFP was a kind gift from Dr. Takahiro Furuta (Kyoto University). DNA sequences of Histone 2B (H2B), LEXY, glucocorticoid receptor (GR), membrane-bound ALFA and NanoLuc were derived from (Addgene ID: 2097), (Addgene ID: 72655), (NM_001364180.2), (Goetzke, 2019)[65] and pNL1.1 vector (Promega, Madison, WI, USA). They were synthesized with restriction enzyme sites and inserted into pAAV-EF1α-H2B-BFP-WPRE, pAAV-CAG-FLEX-H2B-GFP-WPRE, pAAV-EF1α-NLS-mCherry-LEXY-WPRE, pAAV-EF1α-mCherry-GR-WPRE, pAAV-EF1α-EGFP-GR-WPRE, pAAV-CAG-ALFA-WPRE and pAAV-CAG-FLEX-NanoLuc-WPRE, respectively. The FLEX switch consists of paired loxP and lox2272 sequences and enables the expression of a gene of interest only when Cre recombinase is functional[33].

**Cell culture and in vitro luciferase assay.** Plasmids encoding EF1α-driven fluorescent target proteins and N- and C-terminal split Cre chimeric variants were transfected by calcium phosphate into HEK293 cells (AAV293 cells purchased from Agilent Technologies, Inc., Santa Clara, CA, USA) along with plasmids encoding pAAV-CAG-FLEX-NanoLuc. Between 100 and 200 ng of total DNA was transfected into single wells of 96-well plates. Cells were ~80–100% confluent at the time of transfection. Cells were harvested 1 day (Figs. 1 and 3) or 2 days (Fig. 3f) later for the Nano-Glo Luciferase Assay System (Promega). All transfections were done at equal plasmid molar ratios. We used a Spark10M multimode microplate reader (TECAN, Männedorf, Switzerland) to detect luminescence. For the experiments for which results are shown in Supplementary Fig. 2, an mCherry-expressing HEK293 cell line was purchased from Applied StemCell (AST-1320). This stable cell line was generated via the integration of mCherry into the TAR-GATT HEK293 Master cell line with the use of a unique integrase and an mCherry control plasmid.

**Adeno-associated virus (AAV) production and purification.** All AAV vectors were produced using the AAV Helper-Free System (Agilent Technologies, Inc., Santa Clara, CA, USA) and purified on the basis of published methods[66]. Briefly, HEK293 cells were transfected with a pAAV vector plasmid that included a gene of

interest, pHelper and pAAV-RC (serotype 9; purchased from Penn Vector Core, Philadelphia, PA, USA) using the standard calcium phosphate method. Three days later, transfected cells were collected and suspended in artificial cerebrospinal fluid (aCSF; 124 mM NaCl, 3 mM KCl, 26 mM NaHCO₃, 2 mM CaCl₂, 1 mM MgSO₄, 1.25 mM KH₂PO₄, and 10 mM D-Glucose). After 4 freeze-thaw cycles, the cell lysate was treated with benzonase nuclease (Merck, Darmstadt, Germany) at 45 °C for 15 min and centrifuged 2 times at 16,000 g for 10 min. The supernatant was used as the virus-containing solution. To measure the titer of purified virus, the supernatant was dissolved in artificial CSF. Digital PCR was performed to measure the viral titer using TaqMan MGB probes and the following primer pairs: woodchuck hepatitis virus posttranscriptional regulatory element (WPRE): 5'-VIC-CTGCTTTAATGCCTTTGTAT-MGB-3', Forward: 5'-TGCTCCTTTTACGC-TATGTGGATA-3', Reverse: 5'-CATAAAGAGACAGCAACCAGGATTT-3'; human growth hormone polyA: 5'-FAM-CACAATCTTGGCTCACTG-MGB-3', Forward: 5'-GGGTCTATTGGGAACCAAGCT-3', Reverse: 5'-GGCTGAGG-CAGGAGAATCG-3'. The AAV vector was stored at −80 °C in small aliquots until the day of the experiment.

**Stereotaxic AAV injection.** Surgeries for AAV injections were conducted using a stereotaxic instrument. In Cre-DOR C57BL/6 J mouse experiments (Fig. 4), 600 nl of a mixture of AAV9-EF1α-NCre-MBP6-WPRE (1 × 10¹² vg/ml), AAV9-EF1α-CCre-MBP1-WPRE (1 × 10¹² vg/ml), AAV9-CAG-FLEX-nlsGFP-WPRE (1 × 10¹² vg/ml), and AAV9-EF1α-mRFP1-WPRE (5 × 10¹⁰ vg/ml) or AAV9-EF1α-mRuby-WPRE (5 × 10¹⁰ vg/ml) was injected in the right hemisphere M1 cortex (from bregma +0.7 mm, lateral +1.7 mm, ventral −0.8 mm) of 10-week-old male C57BL/6 J mice. In *Esr2*-mRFP1 mouse experiments (Fig. 5), 1 μl of a mixture of AAV9-EF1α-NCre-MBP6-WPRE (final concentration in the mixture: 6 × 10¹² vg/ml), AAV9-EF1α-CCre-MBP1-WPRE (6 × 10¹² vg/ml) and AAV9-CAG-FLEX-palGFP-WPRE (6 × 10¹² vg/ml) was injected in the paraventricular nucleus of the hypothalamus (PVN) (from bregma −0.9 mm, lateral 0.2 mm, ventral −4.3 mm) of 12–14-week-old male and female *Esr2*-mRFP1 transgenic or control wild-type mice. In *Grpr*-mRFP1 rat experiments (Fig. 6), 1 μl of a mixture of AAV9-EF1α-NCre-MBP6-WPRE (final concentration in the mixture: 6 × 10¹² vg/ml), AAV9-EF1α-CCre-MBP1-WPRE (6 × 10¹² vg/ml) and AAV9-CAG-FLEX-palGFP-WPRE (6 × 10¹² vg/ml) was injected in the medial amygdala area (from bregma −3.0 mm, lateral ±3.5 mm, ventral −9.0 mm) of 10–15-week-old female *Grpr*-mRFP1 transgenic or control wild-type rats. Rat brain atlas figures were derived from Swanson, L.W. (2004) Brain maps: structure of the rat brain, 3rd edition (Creative Commons Attribution-NonCommercial 4.0 International License).

**Retrograde tracing using retrobeads.** Surgeries for injections of retrobeads were conducted using a stereotaxic instrument. In *Grpr*-mRFP1 rat experiments (Fig. 5), 300 nl of Green Retrobeads™ IX (Lumafluor, Durham, NC) was injected in the BSTp (from bregma −1.1 mm, lateral 0.8 mm, ventral −7.4 mm) of 9–10-week-old female *Grpr*-mRFP1 transgenic rats. The rats were perfused 1 week after injection.

**Immunohistochemistry and fluorescence microscopy.** Three to four weeks after virus injections, mice were deeply anesthetized by Avertin and transcardially perfused with heparinized saline (20 U/ml) followed by 4% paraformaldehyde in 0.1 M phosphate buffer (pH, 7.4). Brains were removed, post-fixed in 4% paraformaldehyde solution overnight, and transferred to 30% sucrose solution in 0.1 M phosphate buffer (PB) until they sank. Series of 40-μm-thick (Fig. 4) or 30-μm-thick (Figs. 5 and 6) sections were obtained with a cryostat (CryoStar NX70; Thermo Fisher Scientific, Waltham, MA, USA). For staining, coronal brain sections were immersed in a blocking buffer (10% goat or donkey serum and 0.3% Triton-X in 0.1 M PB) and then incubated with primary antibodies at 4 °C overnight. The sections were washed with the blocking buffer and then incubated with secondary antibodies for 1 h at RT. The brain sections were mounted and examined with a fluorescence microscope (IX73, Olympus, Tokyo, Japan). Primary antibodies and secondary antibodies were diluted in the blocking buffer as follows: anti-GFP (RRID: AB_591819, Medical & Biological Laboratories, Tokyo, Japan) at 1:1000 and Alexa Fluor 488 goat anti-rabbit IgG (diluted 1:1000; 1-day incubation at 4 °C; A11034; Thermo Fisher Scientific). For the images shown in Supplementary Fig. 2, we used rat anti-GFP (RRID: AB_2314545, Nacalai tesque, Kyoto, Japan) at 1:2000, rabbit anti-ALFA (NanoTag Biotechnologies, Göttingen, Germany) at 1: 1000, Alexa Fluor 488 donkey anti-rat IgG (A21208; Thermo Fisher Scientific) at 1:1000 and Alexa Fluor 647 donkey anti-rabbit IgG (ab150075; abcam) at 1:1000. For the images shown in Figs. 5 and 6, we used chicken anti-GFP (RRID: AB_1537403, Rockland Immunochemicals, Limerick, PA) at 1:2000, rabbit anti-DsRed (RRID: AB_10013483, Takara Bio, Japan) at 1: 1000, Alexa Fluor 488 goat anti-chicken IgY (diluted 1:1000; 103-545-155; Jackson ImmunoResearch Laboratories, West Grove, PA) and Alexa Fluor 555 goat anti-rabbit IgG (diluted 1:1000; A21428; Thermo Fisher Scientific). We manually counted fluorescence-positive cells using NIH ImageJ software and calculated the relative percentage of fluorescence-positive cells. For counting in Fig. 4, we analyzed one out of every four coronal brain slices.

**Statistics and reproducibility.** Statistical analyses were performed using GraphPad Prism 9 for Windows (GraphPad Software, San Diego, CA). Simple comparisons of the means and SEM were performed by Student's *t*-test. Multiple

comparisons of the means and SEM were performed by one-way ANOVA followed by Tukey's test. A $P$ value of less than 0.05 was considered significant in these analyses.

**Reporting summary**. Further information on research design is available in the Nature Research Reporting Summary linked to this article.

## Data availability

All the data supporting this study are available from the corresponding authors upon reasonable request.

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

## Acknowledgements

This work was supported by KAKENHI grants (16K08527, 16H01488, 17H06061, 19K06910, 22K06487 to A.I.; 17H04026, 17K19636, 25118008 to T.O.). A.I. was also supported by Smoking Research Foundation and a grant from Japan Prize Foundation. We thank Ms. Junko Kato for technical assistance. We thank Dr. T. Furuta (Kyoto University) for the generous gift of a vector coding palGFP. This work was also supported by the program for Brain Mapping by Integrated Neurotechnologies for Disease Studies (Brain/MINDS) from Japan Agency for Medical Research and Development, AMED, under the grant number JP19dm0207057.

## Author contributions

A.I. and T.O. designed the experiments; A.I., S. M., H.M., R.K., R.N., K.T., and H.S. performed the experiments; A.I., S. M., H.M., R.K., R.N., K.T., and H.S. contributed to the analysis and interpretation of data; A.I. and T.O. wrote the manuscript.

## Competing interests

The authors declare no competing interests.
