## [Peer Review File · Communications Biology]

Reviewers' comments:

Reviewer #1 (Remarks to the Author):

The authors present a novel tool box allowing to use mice lines expressing RFP fusion proteins to control the expression of Cre recombinase using nanobody-based approaches.

The authors describe in detail an extremely useful approach, although similar methods based on GFP have previously been established and published. The manuscript is easy to read and the approach is clearly outlined. Since I am not a neurobiologist, the significance of the reported results should be looked at by an expert. The authors also show well-chosen control experiments for target-dependent Cre activation.

Also, I am not a mouse expert, so questions that might arise due to the specificity of expression of the tested components in various regions of the brain need to be evaluated by an expert.

Minor point

First report of protein degradation was Caussinus et al., 2012. It is not obvious why another reference is cited.

Figure 3 not entirely clear (GR experiment)

Reviewer #2 (Remarks to the Author):

Transgenic animal lines expressing fluorescent proteins (FP) have been used to inquire gene expression patterns, cellular morphology of specific cell types, and localization of specific proteins. The authors devised a method to change mCherry (mRFP) reporter lines to the different reporters using a new nanobody-based split Cre (Cre-DOR) recombination system by modifying the previous GFP-dependent split Cre strategy (Cre-DOG; Tang et al., 2015). The authors also demonstrated that this method worked in different neurons in transgenic mice and rat lines using recombinant AAVs. Furthermore, the authors confirmed the previous anatomical information and suggest their future investigation. Finally, the authors show that the efficiency of Cre-DOR is influenced by intracellular localization of the prey RFP, raising the possibility that it may not work for membrane-bound RFPs in some cases. The demonstration showing the orthogonality of recombination activity between Cre-DOR and Flp-DOG is an exciting direction. Most experiments were carefully designed and convincingly demonstrated.

However, there are two major limitations.

1) It does not look that this system is widely useful at this moment. This is because transgenic lines expressing mCherry are not as common as GFP or tdTomato lines as the authors discussed in line 519 (Sasaki et al. 2009 did not use mCherry). One potential advantage of this Cre-DOR method would be to reveal the morphology of individual projections or cells by sparsely labeling a small number of cells. However, the authors essentially show bulk and dense labeling, not sparse labeling. If possible, it would be nice to have such a picture.

2) As seen in several experiments (Figure 2 and Figure 4), this system shows some background. Although the authors described this problem in the manuscript, they do not mention it explicitly. As this work focuses on the method, this caution should be stated clearly. If possible, comparison to the previous GFP-dependent system (Cre-DOG) would be nice (eg, less leaky or more background).

Minor points

3) Abstract, Line 53. $93.5 \pm 0.6\%$ of GFP-positive cells. Is it necessary to include this raw value in the abstract? It does not sound that this information is scientifically valuable because it is from some arbitrary experiments.

4) Line 109. I think that most nanobodies are actually from llama, not camels (including RFP

nanobodies, Fridy et al., 2014).

5) Line 262. PBS? phosphate buffer?

6) As to Figure 1, I wonder how this experiment is reproducible, or meaningful. For example, is N8C1 always better than N1C8? How about the subtle difference? Can we think about new constructs based on this heat map? It is all right to show the result as a screening experiment. However, authors might want to explain how readers can interpret this heat map.

7) Figure 4F. mRuby does not bind to either N-Cre or C-Cre. Thus, this drawing is confusing. It is better to separate them.

8) Figure 5C. It is not clear how cells expressing mRFP1 and palGFP overlap. In addition, it looks that several large GFP+ aggregates are present in RFP-negative areas. Are these GFP+ structures varicosities or somata? Is it possible to add high-power pictures (like Figure 6B)?

Reviewer #3 (Remarks to the Author):

In the presented manuscript entitled "Nanobody-based RFP-dependent Cre recombinase for selective anterograde tracing in RFP-expressing transgenic animals" Inutsuka and colleagues describe the development of a split Cre recombinase for selective labelling of cells expressing monomeric red-fluorescent proteins (mRFPs) and the application of their system for neuronal tracing in the central nervous system of transgenic mice and rats. The overall approach is based on a previous study wherein the activity of a split recombinase was restored upon binding to the green fluorescent protein GFP (Tang et al. *Nat Neurosci* 18:1334-1341). In the current study, the authors tested the combination of eight peptides binding mRFP (either nanobodies or designed ankyrin repeat proteins) fused to either the N-terminal or C-terminal part of the split Cre in cell culture experiments and identified several combinations showing specific binding to mRFPs (either mCherry or mRFP1), but not to dimeric RFPs (tdTomato) or distinct fluorescent proteins (mRuby). It was further shown that the efficiency of Cre-mediated recombination depended on the intracellular localization of the mRFPs with highest recombination efficiency when using nuclear targeted RFP. Based on its high efficiency and specificity one pair of N/C-terminal Cre fragments, named Cre-DOR, was chosen for testing recombination in mouse and rat brains *in vivo*. Being co-transduced using adeno-associated viruses a specificity of ~ 93% was observed (93% of cells expressing flexed GFP were RFP-positive). The concept was then transferred to transgenic animals with cell-type specific RFP expression and used for tracking neuronal projections between different brain areas.

The manuscript is exceptionally clearly written and all experimental procedures and results are described in detail. In line, there is adequate use of references supporting the different parts of the manuscript. As the experimental approach is similar to the one previously described for GFP (see above), the ground-breaking nature of the presented work is limited. There are, however, interesting aspects, which deepen our general understanding of restoration of Cre-activity, such as the different responses observed towards monomeric and dimeric RFP variants, and the importance of intracellular localization for effective Cre recombination. The developed tool by itself might be useful for specific applications, requiring transgenic animals already expressing mRFPs in the target cells. The authors claim that the presented approach should be transferable to link Cre recombinase activity to the expression (or nuclear localization) of endogenous proteins, a very interesting outlook (such tools could be of great interest to many scientists).

Detailed feedback:

1) For the experiments in cell culture and in wildtype mice, there might be an over-estimation of the specificity of the approach as both co-transfection and co-transduction with the same virus may lead to preferential expression of the different components in the same cells/cell population. In cell culture experiments, this bias could be removed by using different means of gene delivery for Cre-DOR, the reporter gene and/or the RFP, e.g. when combining viral delivery with transfection or stable cell lines. For the experiments using transgenic mice, such bias should not

be present. However, while it was shown that 24% of RFP-positive cells expressed GFP in the mouse model, the opposite (RFP-positive cells out of GFP-positive cells) was not quantified. Could you please provide this crucial quantification or is it not possible due to different intracellular localization of the used FPs? Some data is also included in Supplementary Figure 1, but it is not described sufficiently in the main text. In line, would it be possible to show and quantify co-expression of RFP and GFP within individual neurons for the rat experiments?

2) In the discussion (line 507) use of LINus is stated. However, strictly speaking only the LEXY system was used for light-induced relocalization studies.

3) When discussing dimers, it is stated that "the same recognition site should be exposed at two locations in one dimer" (line 566). This would not be true in case the dimer does not show a truly symmetrical arrangement, please rephrase accordingly.

4) The possibility of building light-activated Cre systems based on dimerization is mentioned (lines 572 and 573). Many light-activated Cre systems have been reported, often building on a split Cre approach. Please include the most important references and compare them to the here presented approach.

5) Figure 1: Please introduce all abbreviations, e.g. N-VMA, C-VMA and 10G in 1A.

6) Figure 1B/C: Please label the chosen combination in the grid. You may also want to discuss certain overall patterns, such as preferential binding of MBP 4 and 6. It would also be interesting to discuss why certain binding domains work better when being coupled to the N-terminus/C-terminus? While there seems to be a correlation between pair a/b to the performance of pair b/a, the exact position of the MBP seems to affect the overall outcome (efficiency and specificity of Cre recombination).

7) Figure 3: The abbreviations Cre-DOM and Cre-DOT are only used in this specific figure.

8) Figure 5C/ 6B: Please provide better quality images, if possible with higher resolution. At the current resolution, it is difficult to distinguish individual cells to assess co-labelling with the different FPs.

9) Supplementary Figure 2: This is a very interesting experiment. Please describe in more detail in the main text.

Responses to Reviewers (COMMSBIO-22-0007A)

Reviewer #1 (Remarks to the Author):

The authors present a novel tool box allowing to use mice lines expressing RFP fusion proteins to control the expression of Cre recombinase using nanobody-based approaches.

The authors describe in detail an extremely useful approach, although similar methods based on GFP have previously been established and published. The manuscript is easy to read and the approach is clearly outlined. Since I am not a neurobiologist, the significance of the reported results should be looked at by an expert. The authors also show well-chosen control experiments for target-dependent Cre activation.

Also, I am not a mouse expert, so questions that might arise due to the specificity of expression of the tested components in various regions of the brain need to be evaluated by an expert.

Thank you for your evaluation of our method. Although the emphasis of our manuscript is on *in vivo* usability in transgenic animals, we also focus on the molecular mechanisms *in vitro*.

Minor point

First report of protein degradation was Caussinus et al., 2012. It is not obvious why another reference is cited.

Thank you for your useful comment. We added (Caussinus, 2011-Nat Struct Mol Biol) in the discussion as below.

“Specific recognition of target proteins by nanobodies can be utilized for not only visualization of target proteins^{22, 23, 24} but also protein degradation^{25, 26},” (lines 84 – 85)

Figure 3 not entirely clear (GR experiment)

To clarify the meaning of Figure 3, we added the following sentences in the results section.

“If the Cre-DOR system is dependent on nuclear localization of target RFP proteins, it suggests a limitation of Cre-DOR application for selective gene expression in cells expressing membrane proteins fused with RFPs such as channelrhodopsin 2-mCherry or hM3Dq-mCherry in transgenic animals. On the other hand, it also suggests that the Cre-DOR system can be used for clarification of nuclear translocation of proteins such as nuclear receptors.” (lines 385 - 391)

Reviewer #2 (Remarks to the Author):

Transgenic animal lines expressing fluorescent proteins (FP) have been used to inquire gene expression patterns, cellular morphology of specific cell types, and localization of specific proteins. The authors devised a method to change mCherry (mRFP) reporter lines to the different reporters using a new nanobody-based split Cre (Cre-DOR) recombination system by modifying the previous GFP-dependent splitCre strategy (Cre-DOG; Tang et al., 2015). The authors also demonstrated that this method worked in different neurons in transgenic mice and rat lines using recombinant AAVs. Furthermore, the authors confirmed the previous anatomical information and suggest their future investigation. Finally, the authors show that the efficiency of Cre-DOR is influenced by intracellular localization of the prey RFP, raising the possibility that it may not work for membrane-bound RFPs in some cases. The demonstration showing the orthogonality of recombination activity between Cre-DOR and Flp-DOG is an exciting direction. Most experiments were carefully designed and convincingly demonstrated.

Thank you for your evaluation of the research design of our experiments. We greatly appreciate your insightful and scrupulous discussions in the review. We performed several experiments to address your concerns. We hope you will be satisfied with the revised manuscript.

However, there are two major limitations.

1) It does not look that this system is widely useful at this moment. This is because transgenic lines expressing mCherry are not as common as GFP or tdTomato lines as the authors discussed in line 519 (Sasaki et al. 2009 did not use mCherry). One potential advantage of this Cre-DOR method would be to reveal the morphology of individual projections or cells by sparsely labeling a small number of cells. However, the authors essentially show bulk and dense labeling, not sparse labeling. If possible, it would be nice to have such a picture.

According to the reviewer's advice, we performed several experiments to verify the usability of our methods and to confirm the rigidity of our experiments. First, we performed sparse labeling by Cre-DOR^{N6C1} in Grpr-mRFP1 transgenic rats. Vectors for Cre- Cre-DOR^{N6C1} and FLEX-hrGFP (3 x 10¹¹ vg/ml each, 3 µl) were unilaterally injected into the preoptic area of 10–15-week-old male Grpr-mRFP1 transgenic rats (**Supplementary Figure 6**). Although the expression of hrGFP was sparse, 92.5% of the hrGFP-expressing neurons were confirmed to be mRFP1-expressing neurons. It is likely that Grpr-positive neurons in the preoptic area are peptidergic neurons from their

neuroanatomical features. Numerous axonal varicosities and aspiny dendrites are features of many peptidergic neurons. Indeed, hrGFP-labeled neurons appear to be aspiny, and we found numerous axonal varicosities labeled with hrGFP in the preoptic area (**Supplementary Figure 5, arrowheads**) or more caudal and ventrolateral region from the injection site of the preoptic area.

We removed (Sasaki, 2009) from the references according to the reviewer's advice.

2) As seen in several experiments (Figure 2 and Figure 4), this system shows some background. Although the authors described this problem in the manuscript, they do not mention it explicitly. As this work focuses on the method, this caution should be stated clearly. If possible, comparison to the previous GFP-dependent system (Cre-DOG) would be nice (eg, less leaky or more background).

Thank you for your important suggestion. According to your advice, we added clear descriptions of the background expression of Cre-DOR as below.

“In vitro experiments using HEK293 cells (Figure 2, Supplementary Figures 1, 2) suggest that approximately 80% of transfected cells can induce RFP-dependent recombination by Cre-DOR^{N6C1}, while approximately 8% of transfected cells can induce RFP-independent recombination. Therefore, we recommend using Cre-DOR^{N5C4} when leakage expression should be strictly prevented, while leakage expression in vivo is substantially improved in experiments using AAV vectors (Figures 4, 5, 6, Supplementary Figures 3, 4).” (lines 574 - 580)

According to the reviewer's advice, we also added new data on direct comparison to Cre-DOG in **Supplementary Figure 1**. We found that Cre-DOR^{N6C1} is more effective than Cre-DOG, while Cre-DOR^{N6C1} has more background than Cre-DOG. We also found that Cre-DOR^{N5C4} is more selective than Cre-DOG, while Cre-DOR^{N5C4} is less effective than Cre-DOG. These findings show that our Cre-DOR systems are comparable with the previous Cre-DOG system under conditions of adequate

selection of binding pairs.

Minor points

3) Abstract, Line 53. $93.5 \pm 0.6\%$ of GFP-positive cells. Is it necessary to include this raw value in the abstract? It does not sound that this information is scientifically valuable because it is from some arbitrary experiments.

Thank you for your suggestion. We deleted the raw value in the abstract.

4) Line 109. I think that most nanobodies are actually from llama, not camels (including RFP nanobodies, Fridy et al., 2014).

We appreciate your precise description. We corrected the sentence as below.

“Nanobodies are single-chain small antibodies derived from Camelidae such as llamas” (lines 80 - 81)

5) Line 262. PBS? phosphate buffer?

This PBS is incorrect, we mean PB. We corrected the description as below.

*“Brains were removed, post-fixed in 4% paraformaldehyde solution overnight, and transferred to 30% sucrose solution in 0.1 M phosphate buffer (PB) until they sank. Series of 40- μ m-thick (**Figure 4**) or 30- μ m-thick (**Figures 5 and 6**) sections were obtained with a cryostat (CryoStar NX70; Thermo Fisher Scientific, Waltham, MA, USA). For staining, coronal brain sections were immersed in a blocking buffer (10% goat serum and 0.3% Triton-X in 0.1 M PBS)”* (line 227 - 232)

We also corrected the secondary antibody information in Immunohistochemistry. (lines 247 – 248)

6) As to Figure 1, I wonder how this experiment is reproducible, or meaningful. For example, is N8C1 always better than N1C8? How about the subtle difference? Can we think about new constructs based on this heat map? It is all right to show the result as a screening experiment. However, authors might want to explain how readers can interpret this heat map.

Thank you for insightful questions. About reproducibility of the heatmaps, please refer to the left-side bar graphs indicating mean \pm SEM. Although the deviations are large, we think that the heatmap is reliable to predict the efficiency of recombination in HEK293 cells to some extent. In accordance with our estimation, we confirmed that N8C1 is better than N1C8 as below (*not included in the manuscript). In **Figure 2F**, you can also find that N5C4 recognizes mCherry, but much less effectively mRFP1, and this finding is also predictable by heat maps shown in **Figure 1**.

7) Figure 4F. *mRuby* does not bind to either *N-Cre* or *C-Cre*. Thus, this drawing is confusing. It is better to separate them.

According to the reviewer's advice, we separated them to avoid misunderstanding. For your information, we intended to show virus particles as hexagonals in Figure 4F.

8) Figure 5C. It is not clear how cells expressing *mRFP1* and *palGFP* overlap. In addition, it looks that several large *GFP+* aggregates are present in *RFP-*negative areas. Are these *GFP+* structures varicosities or somata? Is it possible to add high-power pictures (like Figure 6B)?

We appreciate the reviewer's insightful observation. We think that large *GFP+* aggregates in *RFP-*negative areas include both axonal varicosities and cell somata. It is very important to note two possibilities. (1) *mRFP1* expression under control of the *Esr2* promoter can be up- or down-regulated transiently during *Cre-DOR* experiments. (2) A very weak *mRFP1* signal that cannot be detected by immunohistochemistry might induce recombination by *Cre-DOR*. Because we performed negative control experiments using *mRFP1*-negative littermates and found very few *GFP* signals in the injected brain regions (**Figure 5C, 5E**). These results combined with similar results shown in Figure 6 convinced us that *RFP*-negative *GFP* signals in transgenic animals are mainly derived from the above-stated two reasons.

Reviewer #3 (Remarks to the Author):

In the presented manuscript entitled “Nanobody-based RFP-dependent Cre recombinase for selective anterograde tracing in RFP-expressing transgenic animals” Inutsuka and colleagues describe the development of a split Cre recombinase for selective labelling of cells expressing monomeric red-fluorescent proteins (mRFPs) and the application of their system for neuronal tracing in the central nervous system of transgenic mice and rats. The overall approach is based on a previous study wherein the activity of a split recombinase was restored upon binding to the green fluorescent protein GFP (Tang et al. Nat Neurosci 18:1334-1341). In the current study, the authors tested the combination of eight peptides binding mRFP (either nanobodies or designed ankyrin repeat proteins) fused to either the N-terminal or C-terminal part of the split Cre in cell culture experiments and identified several combinations showing specific binding to mRFPs (either mCherry or mRFP1), but not to dimeric RFPs (tdTomato) or distinct fluorescent proteins (mRuby). It was further shown that the efficiency of Cre-mediated recombination depended on the intracellular localization of the mRFPs with highest recombination efficiency when using nuclear targeted RFP. Based on its high efficiency and specificity one pair of N/C-terminal Cre fragments, named Cre-DOR, was chosen for testing recombination in mouse and rat brains in vivo. Being co-transduced using adeno-associated viruses a specificity of ~ 93% was observed (93% of cells expressing flexed GFP were RFP-positive). The concept was then transferred to transgenic animals with cell-type specific RFP expression and used for tracking neuronal projections between different brain areas.

The manuscript is exceptionally clearly written and all experimental procedures and results are described in detail. In line, there is adequate use of references supporting the different parts of the manuscript. As the experimental approach is similar to the one previously described for GFP (see above), the ground-breaking nature of the presented work is limited. There are, however, interesting aspects, which deepen our general understanding of restoration of Cre-activity, such as the different responses observed towards monomeric and dimeric RFP variants, and the importance of intracellular localization for effective Cre recombination. The developed tool by itself might be useful for specific applications, requiring transgenic animals already expressing mRFPs in the target cells. The authors claim that the presented approach should be transferable to link Cre recombinase activity to the expression (or nuclear localization) of endogenous proteins, a very interesting outlook (such tools could be of great interest to many scientists).

Thank you for your kind valuation of our work. In particular, we agree that Cre dependent on endogenous proteins will be great interest to many scientists. Although there are only limited numbers of nanobodies for endogenous proteins, the situation has been dramatically improved. A

recent report showed that machine learning using only structural information can produce specific small-sized binding proteins (Cao et al., 2022). We also strengthened our discussion on this part by adding the description in a previous report (Liu et al., 2017). In that paper, reports reunion of split-luciferase combined with nanobodies for endogenous CD38 protein is described. These findings suggest that our method can be transferred to endogenous proteins when a sufficient amount of candidate specific nanobodies or small binding proteins is available. We added some discussion as below.

“A recent paper also showed that machine learning using only structural information of a target protein can produce specific small-sized specific binding proteins⁶⁸. An endogenous protein can be used as a target protein for reunion of split-luciferase fused with specific nanobodies⁶⁹.” (lines 641 – 645)

“Taken together, our results suggest a potential for AAV vectors for target protein-based genetic manipulation based on AAV vectors.” (lines 650 – 652)

Detailed feedback:

1) For the experiments in cell culture and in wildtype mice, there might be an over-estimation of the specificity of the approach as both co-transfection and co-transduction with the same virus may lead to preferential expression of the different components in the same cells/cell population. In cell culture experiments, this bias could be removed by using different means of gene delivery for Cre-DOR, the reporter gene and/or the RFP, e.g. when combining viral delivery with transfection or stable cell lines. For the experiments using transgenic mice, such bias should not be present. However, while it was shown that 24% of RFP-positive cells expressed GFP in the mouse model, the opposite (RFP-positive cells out of GFP-positive cells) was not quantified. Could you please provide this crucial quantification or is it not possible due to different intracellular localization of the used FPs? Some data is also included in Supplementary Figure 1, but it is not described sufficiently in the main text. In line, would it be possible to show and quantify co-expression of RFP and GFP within individual neurons for the rat experiments?

We appreciate your precise comments on co-transfection. In HEK293 experiments, we recognize that co-transfection affects the calculation of RFP/GFP percentages and we stated that “In all cases, the percentages of RFP-positive cells in GFP-positive cells were higher than 90% possibly because of the transfection method (mCherry: $98.9 \pm 0.3\%$, mRFP1: $98.3 \pm 0.3\%$, mRuby: $91.0 \pm 1.9\%$, mRFP1 Δ CCre: $94.4 \pm 3.7\%$, mRFP1 Δ NCre: $88.5 \pm 5.6\%$).” Therefore, we did not describe the RFP/GFP percentages as reliable information in HEK293 experiments. In order to make the situation clear, we performed experiments using an mCherry-expressing stable line (**Supplementary Figure**

2) as below.

This stable line expresses mCherry in more than 90% of the cells (**Supplementary Figure 2A**). Four kinds of plasmids including N-Cre-MBP6, C-Cre-MBP1, membrane-bound ALFA tag, and FLEX-

nlsgFP were co-transfected into HEK293 cells (**Supplementary Figure 2B**). Membrane-bound ALFA tag was transfected to confirm the transfected cells by immunocytochemistry using anti-ALFA tag antibody. Quantitative cell counting of fluorescent images showed that $79.8 \pm 3.3\%$ of the ALFA-positive cells were GFP-positive in the mCherry-expressing stable cell line and that $7.5 \pm 2.1\%$ of the ALFA-positive cells were GFP-positive in the control cell line ($n = 4$ each) (**Supplementary Figure 2C, 2D**). Considering **Figure 2**, these results suggest that approximately 80% of transfected cells can induce RFP-dependent recombination by Cre-DOR^{N6C1}, while approximately 8% of transfected cells can induce RFP-independent recombination.

We also performed experiments with two modifications for reducing the co-infection of virus vectors: (1) change in serotype of AAV vectors and (2) segregation of virus injection.

AAV-EF1a-mRFP1-WPRE or AAV-EF1a-mRuby-WPRE (200 μ l; 2×10^{11} vg/mouse) was injected systemically (**Supplementary Figure 4A**). They are packaged by AAV PHPeb and can infect through the blood-brain barrier. Within the same day, 600 μ l of a mixture of virus vectors for Cre-DOR^{N6C1} and FLEX-nlsGFP (1×10^{12} vg/ml each) was unilaterally injected into the right side M1 cortex of wild-type 10-week-old male mice (**Supplementary Figure 4A**). Four weeks after injection, mice were sacrificed for immunohistochemistry and brain slices were stained with anti-GFP. Clear

expression of nlsGFP at the injected site in the M1 cortex was observed (**Supplementary Figure 4B**). While we observed comparative amounts of mRuby-expressing neurons in the M1 cortex, we found only sparse nlsGFP-expressing neurons in the same area. Quantitative cell counting of fluorescent images showed that $49.9 \pm 3.9\%$ of the mRFP1-positive cells were GFP-positive and that $93.3 \pm 0.4\%$ of the GFP-positive cells were mRFP1-positive ($n = 4$ each) (**Supplementary Figure 4C**). Quantitative cell counting of nlsGFP and mRuby-positive cells showed a clear difference between Cre-DOR^{N6C1}+ mRFP1-injected mice and Cre-DOR^{N6C1} + mRuby-injected mice. The cell counting showed that $1.7 \pm 0.7\%$ of the mRuby-positive cells were GFP-positive and that $10.5 \pm 3.6\%$ of the GFP-positive cells were mRuby-positive ($n = 4$ each) (**Supplementary Figure 4D**). The recombination efficiency of Cre-DOR^{N6C1} + mRFP1 was 29.9-times higher than that of Cre-DOR^{N6C1} + mRuby in these experiments. All of these data confirmed *in vivo* specificity of the Cre-DOR^{N6C1} system using AAV vectors.

2) In the discussion (line 507) use of LINus is stated. However, strictly speaking only the LEXY system was used for light-induced relocalization studies.

Thank you for your comment. We corrected the description as below.

“we achieved optical and pharmacological manipulation of Cre-DOR activity by utilizing intracellular translocation of the light-inducible nuclear export system (LEXY) and glucocorticoid receptor (Figure 3).” (lines 552 - 554)

3) When discussing dimers, it is stated that “the same recognition site should be exposed at two locations in one dimer” (line 566). This would not be true in case the dimer does not show a truly symmetrical arrangement, please rephrase accordingly.

Thank you for your fine discussion. It is true that the recognition site is not exposed in two loci if the dimer is asymmetrical, or recognition sites are used for binding. We rephrased the sentence as below.

“the same recognition site can be exposed at two locations in one dimer at the same time.” (lines 615 - 616)

4) The possibility of building light-activated Cre systems based on dimerization is mentioned (lines 572 and 573). Many light-activated Cre systems have been reported, often building on a split Cre approach. Please include the most important references and compare them to the here presented approach.

Thank you for your constructive suggestion. We think that such a method might be useful for simultaneous achievement of fluorescence imaging and optical genetic manipulation, although there are several reported light-activated Cre systems such as PA-Cre as the reviewer pointed out. However, we decided to remove this discussion because of the word limitation of Communications Biology (Main text: no more than 5,000 words, Our present main text: 4,999 words).

5) Figure 1: Please introduce all abbreviations, e.g. N-VMA, C-VMA and 10G in 1A.

We added full spellings of abbreviations for N-VMA, C-VMA, and 10G in Figure 1A. VMA: vacuolar membrane ATPase subunit, N-VMA: N-terminal portion of VMA, C-VMA: C-terminal portion of VMA, 10G: 10 glycine linker in Figure 1 legend.

6) Figure 1B/C: Please label the chosen combination in the grid. You may also want to discuss certain overall patterns, such as preferential binding of MBP 4 and 6. It would also be interesting to discuss why certain binding domains work better when being coupled to the N-terminus/C-terminus? While there seems to be a correlation between pair a/b to the performance of pair b/a, the exact position of the MBP seems to affect the overall outcome (efficiency and specificity of Cre recombination).

We marked the chosen combination in the grids of Figure 1B and 1C using *, † and ‡.

Thank you for your constructive comments on luciferase assay data. We think that the heatmap is reliable to predict the efficiency of recombination in HEK293 cells to some extent. In accordance with our estimation, we confirmed that N8C1 is better than N1C8 as below (*not included in the manuscript). In **Figure 2F**, you can also find that N5C4 recognizes mCherry, but much less effectively mRFP1, and this finding is also predictable by heat maps shown in **Figure 1**. Although this discussion might be interesting for readers, we decided to remove this discussion because of the word limitation of Communications Biology (Main text: no more than 5,000 words, Our present main text: 4,999 words).

We also agree that the relationship between binding pairs and recombination activity should be discussed; however, it is too speculative at present without structural information on binding sites for nanobodies and DARPins. Upcoming structural studies using crystallography or/and rapidly developing prediction tools such as alphaFold will provide rich information for that kind of discussion.

7) Figure 3: The abbreviations *Cre-DOM* and *Cre-DOT* are only used in this specific figure.

We corrected *Cre-DOM* to *Cre-DOR^{N6C1}* and *Cre-DOT* to *Cre-DOR^{N8C8}*.

8) Figure 5C/ 6B: Please provide better quality images, if possible with higher resolution. At the current resolution, it is difficult to distinguish individual cells to assess co-labelling with the different FPs.

We are sorry that we cannot provide better images for Figure 5C and Figure 6B. We instead provide **Supplementary Figure 6** as below.

9) *Supplementary Figure 2: This is a very interesting experiment. Please describe in more detail in the main text.*

Thank you for your encouraging suggestion. We added some description of **Supplementary Figure 3 (previous Supplemental Figure 2)** in the main text as below.

“This result suggests that it might be possible to perform Boolean operation using Cre-DOR and Flp-DOG. For example, if we use pAAV-hSyn Con/Fon hChr2(H134R)-EYFP⁷¹, we can manipulate only GFP- and RFP-expressing cell populations. Taken together, our results suggest a potential for AAV vectors for target protein-based genetic manipulation based on AAV vectors.” (lines 647 - 652)

References

- Cao, L., Coventry, B., Goreshnik, I., Huang, B., Sheffler, W., Park, J.S., Jude, K.M., Markovic, I., Kadam, R.U., Verschueren, K.H.G., et al. (2022). Design of protein-binding proteins from the target structure alone. *Nature* *605*, 551-560. 10.1038/s41586-022-04654-9.
- Liu, J., Zhao, Y.J., Li, W.H., Hou, Y.N., Li, T., Zhao, Z.Y., Fang, C., Li, S.L., and Lee, H.C. (2017). Cytosolic interaction of type III human CD38 with CIB1 modulates cellular cyclic ADP-ribose levels. *Proc Natl Acad Sci U S A* *114*, 8283-8288. 10.1073/pnas.1703718114.

REVIEWERS' COMMENTS:

Reviewer #2 (Remarks to the Author):

This revised version of the manuscript is a great improvement. The results are now refined and organized to boost confidence in the interpretations. The authors are to be highly commended.

One minor comment

In Supplementary Figure 2C, I wonder why the bottom/right panel has the magenda color. It should be a green (GFP) picture although it must be mostly black based on the merge picture.

Reviewer #3 (Remarks to the Author):

Thank you very much for your detailed revision and addition of further data/text. The new data using a stable cell line and different viral injection sites are a valuable addition to the manuscript. I have no further concerns.

Responses to Reviewers (COMMSBIO-22-0007B)

Reviewer #2 (Remarks to the Author):

One minor comment

In Supplementary Figure 2C, I wonder why the bottom/right panel has the magenda color. It should be a green (GFP) picture although it must be mostly black based on the merge picture.

Thank you for the precise check. We found that although the Supplementary Figure 2 inserted in the manuscript or “Responses to reviewers” file is correct, the tif file was incorrect as the reviewer pointed out (a mistake of the link in the original illustrator file). We corrected this mistake in the revision. We also moved scale bar information from the supplemental figures to their legends.